# A Multicover Approach to Neural Networks Sample Complexity

## Abstract

Covering numbers are central to estimating sample complexity. Alas, standard techniques for bounding covering numbers fail in estimating the covering numbers of many classes of neural networks. We introduce a generalization of covers, called *multicovers*, which are covers w.r.t. many metrics simultaneously. Contrary to standard covering numbers, multicovering numbers behave better with the layer-wise structure in neural networks. We utilize this property to recover a recent result of Daniely & Granot (2019) who defined a new notion called Approximate Description Length (ADL) to establish tight bounds on the sample complexity of networks with weights of bounded Frobenius norm. We also show that ADL and multicovering numbers are closely related.

## 1 Introduction

Covering numbers are one of the most basic techniques for bounding the sample complexity of function classes, and can achieve state of the art bounds in various cases. Alas, it is not clear how to estimate covering numbers for function classes of layered architectures, such as neural networks. Indeed, state-of-the-art results still exhibit a polynomial gap between upper and lower sample complexity bounds. This is in contrast to non-layerd function classes, in which the gaps are often logarithmic or even constant.

A major flaw of covering numbers is that it is not clear how to use them inductively on the network's layers. That is, a bound on the covering number for function classes of depth $i$ architectures, is not enough to derive a tight (up to log factors) bound for function classes of depth $i + 1$ architectures.

In this paper, we present a generalization of covering called *multicover*, which overcomes the above barrier, at least in some cases. This allows the derivation of tight bounds on various families of neural networks. In a nutshell, given a set $\mathcal{S}$ of $d \times d$ PSD matrices, an $\epsilon$-multicover of a set $\mathcal{X} \subset \mathbb{R}^d$ is a set $\check{\mathcal{X}}$ that simultaneously forms an $\epsilon$-cover w.r.t. to any metric of the form $d(\mathbf{x}, \mathbf{y}) = \sqrt{(\mathbf{x} - \mathbf{y})^\top R (\mathbf{x} - \mathbf{y})}$ for $R \in \mathcal{S}$. The $\epsilon$-multicovering number of $\mathcal{X}$ is the minimal size of an $\epsilon$-multicover of $\mathcal{X}$. We note that if $\mathcal{S}$ consists of a single matrix $R$, multicover is just a standard cover w.r.t. the norm corresponding to $R$. However, for other sets $\mathcal{S}$ we get a notion of covering that is fundamentally different from standard covering w.r.t. a metric.

We present techniques which allow for layerwise induction. Using these techniques and for the case of $\mathcal{S}$ being the class of PSD matrices with trace at most 1, we show the following. Given a bound on the multicovering number of $\mathcal{X}$, a class $\mathcal{L}$ of $d \times d$ matrices, and a non-linearity $\sigma : \mathbb{R} \to \mathbb{R}$, we derive bounds which are often tight on the multicovering number of

$$\mathcal{L}\mathcal{X} = \{A\mathbf{x} : A \in \mathcal{L}, \mathbf{x} \in \mathcal{X}\} \text{ and } \sigma(\mathcal{X}) = \{(\sigma(x_1), \ldots, \sigma(x_d)) : \mathbf{x} \in \mathcal{X}\}$$

The tools we present allow us to derive nearly tight bounds on the sample complexity of constant depth networks with weights of bounded norm. For instance, assume that the activation is the ReLU-like softplus activation $\sigma(x) = \log(1 + e^x)$ and consider the class $\mathcal{N}$ of networks of depth $l$, width $d$, and weight matrices with spectral norm at most $O(1)$ and Frobenius norm at most $R$. This and similar classes have been studied intensively in recent years, because sample complexity bounds on such classes can potentially be sublinear in the number of network parameters, thus shedding light on a main mystery of modern neural networks. We show that if the input distribution is supported in $[-1, 1]^d$ then the sample complexity of $\mathcal{N}$ is $\tilde{O}(dR^2)$ which is sublinear in number of parameters and is tight up to poly-log factors.

As far as we know, despite extensive efforts, such results are not known to be derived via "standard" covering number techniques, or even more generally, via other common techniques such as Radamacher complexity. nevertheless, we note that similar bounds were recently proved (Daniely & Granot, 2019) using a notion called Approximate Description Length (ADL). We show that ADL is closely related to multicover, and in a sense, multicover can be seen as a "dual" approach to ADL. We hope that having both the ADL technique and the multi-covering technique at our disposal will lead to further progress in the future.

Throughout this paper, absent proofs for theorems, lemmas, and claims appear in full form in the appendix.

## 1.1 COVERING NUMBERS AND LAYERWISE INDUCTION

We next give a simple example in which layerwise induction fails to establish tight bounds on covering numbers. We emphasize that the goal of this example is to demonstrate the problem with layerwise induction on covering numbers, but it is not a proof that the approach is doomed to fail in general.

We will consider the class $\mathcal{H}$ of linear classifiers of norm $\leq 1$ over $B_{\sqrt{d}}^d$. That is, $\mathcal{H}$ consists of all functions $h : B_{\sqrt{d}}^d \to \mathbb{R}$ of the form $h(\mathbf{x}) = \mathbf{v}^\top \mathbf{x}$ for $\mathbf{v} \in B_1^d$. It is well known that the sample complexity of $\mathcal{H}$ is $\tilde{O}\left(\frac{d}{\epsilon^2}\right)$. In order to prove this via covering numbers one can show that for any choice of $\mathbf{x}_1, \ldots, \mathbf{x}_m \in B_{\sqrt{d}}^d$ the covering number of

$$\mathcal{X} = \{(h(\mathbf{x}_1), \ldots, h(\mathbf{x}_m)) : h \in \mathcal{H}\}$$

satisfies $\log(N_2(\mathcal{X}_2, \epsilon)) = \tilde{O}\left(\frac{d}{\epsilon^2}\right)$. This can be proved using standard covering number techniques.

For the sake of illustration we will view $\mathcal{H}$ a composition of two function classes, corresponding to a two layer neural network. Fix $\mathbf{u} \in \mathbb{S}^{d-1}$, let $\mathcal{H}_1$ be the class of all functions $h : B_{\sqrt{d}}^d \to B_{\sqrt{d}}^d$ of the form $h(\mathbf{x}) = A\mathbf{x}$ for $A \in M_{d,d}$ for $A$ with $\|A\|_F \leq 1$, and let $\mathcal{H}_2 = \{h_\mathbf{u}\}$. Note that $\mathcal{H} = \mathcal{H}_2 \circ \mathcal{H}_1$. Now fix $\mathbf{x}_1, \ldots, \mathbf{x}_m \in B_{\sqrt{d}}^d$ and $\mathbf{u} \in \mathbb{S}^{d-1}$. Consider the sets $\mathcal{X}_1 = \{(h(\mathbf{x}_1), \ldots, h(\mathbf{x}_m) : h \in \mathcal{H}_1\}$ and $\mathcal{X}_2 = \{(h(\mathbf{y}_1), \ldots, h(\mathbf{y}_m)) : (\mathbf{y}_1, \ldots, \mathbf{y}_m) \in \mathcal{X}_1, \ h \in \mathcal{H}_2\}$

As noted above, $\log(N_2(\mathcal{X}_2, \epsilon)) = \tilde{O}\left(\frac{d}{\epsilon^2}\right)$. Suppose now that we want to prove this in an inductive way. Can we guarantee that $\log(N_2(\mathcal{X}_2, \epsilon)) = \tilde{O}\left(\frac{d}{\epsilon^2}\right)$ via a bound on the $\ell^2$ covering numbers of $\mathcal{X}_1$ without specific assumptions on the structure of $\mathcal{X}_1$? (remember that we want an inductive argument that will work for neural networks, in which case it is not clear what further assumptions we can make)? As Claim 1 below shows, without assumptions beyond the fact that $\mathcal{X}_1 \subset \left(B_{\sqrt{d}}^d\right)^m$, the best bound we can derive is $N_2(\mathcal{X}_2, \epsilon) \leq N_2(\mathcal{X}_1, \epsilon)$. This is not enough as for $\epsilon = 1/4$, Claim 2 below chows that $\log(N_2(\mathcal{X}_1, \epsilon))$ may be as large as $\Omega(d^2)$, thus the best bound we can get is $\log(N_2(\mathcal{X}_2, 1/4)) = O\left(d^2\right)$.

**Claim 1.** *There is a set $\mathcal{X} \subset \left(B_{\sqrt{d}}^d\right)^m$ such that $N_2(\mathcal{X}, \epsilon) = N_2(\mathbf{u}^\top \mathcal{X}, \epsilon)$*

*Proof.* (sketch) It is not hard to verify that for $\mathcal{X} = \left\{\left(a_1\sqrt{d}\mathbf{u}, \ldots, a_m\sqrt{d}\mathbf{u}\right) : a_i \in \{\pm 1\}\right\}$. We have $N_2(\mathcal{X}, \epsilon) = N_2(\{\pm\sqrt{d}\}^m, \epsilon) = N_2(\mathbf{u}^\top \mathcal{X}, \epsilon)$ $\qquad \square$

**Claim 2.** *For $\epsilon \leq 1$, $m = d$ and $\mathbf{x}_i = \sqrt{d}\mathbf{e}_i$ we have $N_2(\mathcal{X}_2\epsilon) = \Omega(d^2)$*

*Proof.* We have $\mathcal{X}_1 = \left\{(\mathbf{a}_1, \ldots, \mathbf{a}_d) : \sum_{i=1}^d \|\mathbf{a}_i\|^2 = d\right\}$. Thus, $N_2(\mathcal{X}_1, \epsilon) = N_2\left(B_{\sqrt{d}}^{d^2}, \sqrt{d}\epsilon\right) = N_2\left(B_1^{d^2}, \epsilon\right) \stackrel{\text{Lemma 2.2}}{=} \Omega(d^2)$ $\qquad \square$

## 2 MULTICOVER

### 2.1 NOTATION

We denote by $\mathbf{x}_1\mathbf{x}_2$ the elementwise product of two vectors $\mathbf{x}_1, \mathbf{x}_2 \in \mathbb{R}^d$. We denote by $\mathbf{e}_1 \ldots, \mathbf{e}_d$ the standard basis of $\mathbb{R}^d$ and by $\{E_{ij}\}_{1 \le i,j \le d}$ the standard basis of the space of $d \times d$ matrices. We will use $\|\cdot\|$ to denote the standard Euclidean norm for vector and the spectral norms for matrices. $\|\cdot\|_F$ will be used for the Frobenius norm of matrices. $B_r^d$ will stand for the Euclidean ball of radius $r$ in $\mathbb{R}^d$. We will use $\lesssim$ to denote inequality up to a constant.

### 2.2 BASIC DEFINITIONS

Denote by $\mathcal{R}^d$ the convex set of $d \times d$ PSD matrices. Let $\mathcal{S} \subset \mathcal{R}^d$, we say that $\mathcal{S}$ is *nice* if $\forall R \in \mathcal{S}$ and $W \in \mathbb{R}^{d \times d}$ with $\|W\| \le 1$ then $W^\top RW \in \mathcal{S}$. We denote by $\mathcal{R}_t^d = \{R \in \mathcal{R}^d : \mathrm{Tr}(R) \le t\}$ the corresponding nice set. We denote the inner product, the norm, and the metric induced by $R \in \mathcal{R}^d$ on $\mathbb{R}^d$ by $\langle \mathbf{x}, \mathbf{y} \rangle_R = \langle \mathbf{x}, R\mathbf{y} \rangle$, $\|\mathbf{x}\|_R = \sqrt{\langle \mathbf{x}, \mathbf{x} \rangle_R}$ and $d_R(\mathbf{x}, \mathbf{y}) = \|\mathbf{x} - \mathbf{y}\|_R$. Fix $\mathcal{X} \subset \mathbb{R}^d$ and let $\varepsilon > 0$. A set $\check{\mathcal{X}} \subset \mathbb{R}^d$ is an $\varepsilon$-*cover* of $\mathcal{X}$ w.r.t. a metric $d$ on $\mathbb{R}^d$ if for every $\mathbf{x} \in \mathcal{X}$ there is $\check{\mathbf{x}} \in \check{\mathcal{X}}$ such that $d(\mathbf{x}, \check{\mathbf{x}}) \le \varepsilon$. A set $\check{\mathcal{X}} \subset \mathbb{R}^d$ is an $\varepsilon$-*multicover* of $\mathcal{X}$ w.r.t. a nice set $\mathcal{S}$ if for any $R \in \mathcal{S}$ and every $\mathbf{x} \in \mathcal{X}$ there is $\check{\mathbf{x}} \in \check{\mathcal{X}}$ such that $\|\mathbf{x} - \check{\mathbf{x}}\|_R \le \sqrt{\mathrm{Tr}(R)}\varepsilon$. Equivalently, for any $R \in \mathcal{S}$, $\check{\mathcal{X}}$ is an $\varepsilon$-cover of $\mathcal{X}$ w.r.t. $d_R$. The $\varepsilon$-*multicovering-number* of $\mathcal{X}$, w.r.t. $\mathcal{S}$, and denoted by $M_{\mathcal{S}}(\mathcal{X}, \varepsilon)$ is the minimal size of an $\varepsilon$-multicover of $\mathcal{X}$ w.r.t. $\mathcal{S}$. Likewise, the $\varepsilon$-*covering-number* of $\mathcal{X}$ w.r.t. a metric $d$, denoted by $N_d(\mathcal{X}, \varepsilon)$, is the minimal size of an $\varepsilon$-cover of $\mathcal{X}$ w.r.t. $d$. We will use $N_p(\mathcal{X}, \varepsilon)$ when the metric is $d(\mathbf{x}, \mathbf{y}) = \|\mathbf{x} - \mathbf{y}\|_p$ and $N_R(\mathcal{X}, \varepsilon)$ when the metric is $d_R$ for $R \in \mathcal{R}^d$.

Note that if $\mathcal{S}$ is a nice set, $A \in \mathcal{S}$ and $B \preceq A$, then w.l.o.g. we may assume $B \in \mathcal{R}$. This is because for every $x \in \mathbb{R}^d$ $x^\top Bx \le x^\top Ax$, thereby $\|x\|_B \le \|x\|_A$. i.e. adding all PSD matrices of lower PSD order to $\mathcal{S}$ keeps the $\varepsilon$-multicover w.r.t. $\mathcal{S}$ valid. On the other hand, adding matrices only adds constraints and therefore cannot decrease the multicovering number. Overall we get $M_{\mathcal{S}}(\mathcal{X}, \varepsilon) = M_{\mathcal{S} \cup \{B\}}(\mathcal{X}, \varepsilon)$ for any $\mathcal{X}$ and $\varepsilon$

We will also use the notion of packing. We say that $\check{\mathcal{X}} \subset \mathcal{X}$ is an $\epsilon$-*packing* of $\mathcal{X}$ w.r.t a metric $d$ on $\mathcal{X}$ if $d(\mathbf{x}, \mathbf{y}) \ge \epsilon$ for any pair of points $\mathbf{x}, \mathbf{y} \in \check{\mathcal{X}}$. We denote by $P_d(\mathcal{X}, \epsilon)$ the maximal size of an $\epsilon$-packing of $\mathcal{X}$. As with covering, we will use $P_p(\mathcal{X}, \varepsilon)$ when the metric is $d(\mathbf{x}, \mathbf{y}) = \|\mathbf{x} - \mathbf{y}\|_p$ and $P_R(\mathcal{X}, \varepsilon)$ when the metric is $d_R$ for $R \in \mathcal{R}^d$. It is well known (e.g. Vershynin (2018)) that

$$P_d(\mathcal{X}, 2\epsilon) \le N_d(\mathcal{X}, \epsilon) \le P_d(\mathcal{X}, \epsilon) \tag{1}$$

### 2.3 SOME PRELIMINARY LEMMAS

**Lemma 2.1.** *Let $X_1, \ldots, X_k$ be independent r.v. with that that are $\sigma$-estimators to $\mu$. Then*

$$\Pr\left(|\mathrm{median}(X_1, \ldots, X_k) - \mu| > r\sigma\right) < \left(\frac{2}{r}\right)^k$$

**Lemma 2.2** (e.g. Vershynin (2018)). *For any $\varepsilon \le M$, $(M/\varepsilon)^d \le N_2(B_M^d, \varepsilon) \le (3M/\varepsilon)^d$*

**Lemma 2.3.** $P_2(\{\pm 1\}^d, d) \ge e^{d/8}$

### 2.4 MULTICOVER AND ESTIMATORS

We say that a random variable $X \in \mathbb{R}^d$ is an $\varepsilon$-*estimator* of $\mathbf{x} \in \mathbb{R}^d$ if for any $\mathbf{u} \in \mathbb{S}^{d-1}$, $\mathbb{E}\langle \mathbf{u}, X - \mathbf{x} \rangle^2 \le \varepsilon^2$. Equivalently, for any $R \in \mathcal{R}^d$, $\mathbb{E}\|X - \mathbf{x}\|_R^2 \le \mathrm{Tr}(R)\varepsilon^2$. We say that $X$ is *unbiased* if $\mathbb{E}X = \mathbf{x}$.

**Lemma 2.4.** *Let $\mathcal{X} \subset \mathbb{R}^d$. A set $\check{\mathcal{X}} \subset \mathbb{R}^d$ is an $\varepsilon$-multicover of $\mathcal{X}$ w.r.t. $\mathcal{R}^d$ if and only if for any $\mathbf{x} \in \mathcal{X}$ there is a random vector $X \in \check{\mathcal{X}}$ that is an $\varepsilon$-estimator of $\mathbf{x}$.*

*Proof.* Write $\check{\mathcal{X}} = \{\mathbf{x}_1, \ldots, \mathbf{x}_T\}$. Suppose that $\check{\mathcal{X}}$ is a $\varepsilon$-multicover and let $\mathbf{x} \in \mathcal{X}$. It is enough to show that there is a r.v. $X$ whose range is $\{\mathbf{x}_1, \ldots, \mathbf{x}_T\}$ such that for any $R \in \mathcal{R}_1^d$, $\mathbb{E}\|X - \mathbf{x}\|_R^2 \le \varepsilon^2$.

Such a r.v. exists if and only if

$$\min_{\lambda \in \Delta^{T-1}} \max_{R \in \mathcal{R}_1^d} \sum_{i=1}^{T} \lambda_i \|\mathbf{x}_i - \mathbf{x}\|_R^2 \leq \varepsilon^2$$

since the objective $\sum_{i=1}^{T} \lambda_i \|\mathbf{x}_i - \mathbf{x}\|_R^2 = \sum_{i=1}^{T} \lambda_i (\mathbf{x}_i - \mathbf{x})^\top R (\mathbf{x}_i - \mathbf{x})$ is bi-linear in $\lambda$ and $R$, and since $\Delta^{T-1}$ and $\mathcal{R}_1^d$ are both convex and compact, we can apply the minmax theorem to conclude that a r.v. $X$ as described above exists if and only if

$$\max_{R \in \mathcal{R}_1^d} \min_{\lambda \in \Delta^{T-1}} \sum_{i=1}^{T} \lambda_i \|\mathbf{x}_i - \mathbf{x}\|_R^2 \leq \varepsilon^2$$

which is equivalent to

$$\max_{R \in \mathcal{R}_1^d} \min_{i \in [T]} \|\mathbf{x}_i - \mathbf{x}\|_R \leq \varepsilon$$

Which is indeed the case as $\check{\mathcal{X}}$ is an $\varepsilon$-multicover on $\mathcal{X}$.

Suppose now that for any $\mathbf{x} \in \mathcal{X}$ there is a r.v. $X$ whose range is $\check{\mathcal{X}}$ such that for any $R \in \mathcal{R}_1^d$, $\mathbb{E}\|X - \mathbf{x}\|_R^2 \leq \varepsilon^2$. This implies that for any $\mathbf{x} \in \mathcal{X}$ and any $R \in \mathcal{R}_1^d$ there is $\check{\mathbf{x}} \in \check{\mathcal{X}}$ such that $\|\check{\mathbf{x}} - \mathbf{x}\|_R \leq \varepsilon$. This implies that $\check{\mathcal{X}}$ is an $\varepsilon$-multicover of $\mathcal{X}$. □

## 2.5 THE MULTICOVERING-NUMBER OF AN EUCLIDEAN BALL

**Lemma 2.5.** *For the ball $B_M^d = \left\{ \mathbf{x} \in \mathbb{R}^d \mid \|\mathbf{x}\|_2 \leq M \right\}$ and $\varepsilon \leq M$ we have*

$$2^{\min(d, \lfloor (M/2\varepsilon)^2 \rfloor)} \leq M_{\mathcal{R}_1^d}(B_M^d, \varepsilon) \leq \min\left( (4d^2 \lceil M \rceil + 6d)^{\lceil \frac{2M^2 + \frac{1}{4}}{\varepsilon^2} \rceil}, (3M/\varepsilon)^d \right)$$

The idea behind the proof is constructing a sparse covering set by picking a convex hull that covers the ball, then using averaging to make the sparse cover $k$-sparse, in the spirit of Maury's lemma (Pisier, 1980-1981).

## 2.6 MULTICOVER CALCULUS

**Lemma 2.6.** *Let $\mathcal{S} \subset \mathcal{R}^d$ be a nice set, then:*

1. *For $\mathcal{X} \subset \mathbb{R}^{d_1}$ and a $d_2 \times d_1$ matrix $A$ we have $M_{\mathcal{S}}(A\mathcal{X}, \|A\|\varepsilon) \leq M(\mathcal{X}, \varepsilon)$*

2. *For $\mathcal{X}_1, \ldots, \mathcal{X}_n \subset \mathbb{R}^d$ and $\varepsilon_1, \ldots, \varepsilon_n > 0$ we have $M_{\mathcal{S}}(\sum_{i=1}^{n} \mathcal{X}_i, \sum_{i=1}^{n} \varepsilon_i) \leq \prod_{i=1}^{n} M_{\mathcal{S}}(\mathcal{X}_i, \varepsilon_i)$*

3. *For $\mathcal{X} \subset \mathbb{R}^d, \varepsilon > 0$, orthonormal matrix $U$ and $\mathbf{b} \in \mathbb{R}^d$ we have $M_{\mathcal{S}}(U\mathcal{X} + \mathbf{b}, \varepsilon) = M_{\mathcal{S}}(\mathcal{X}, \varepsilon)$*

4. *For $\mathcal{X}_1, \ldots, \mathcal{X}_n \subset \mathbb{R}^d$ and $\varepsilon > 0$ we have $M_{\mathcal{S}}(\cup_{i=1}^{n} \mathcal{X}_i, \varepsilon) \leq \sum_{i=1}^{n} M_{\mathcal{S}}(\mathcal{X}_i, \varepsilon)$*

5. *For $\mathcal{S} = \mathcal{R}_1^d$ and $\mathcal{X}_i \subset [-M_i, M_i]^d$ and $\varepsilon_1, \ldots, \varepsilon_n > 0$ we have*

$$M_{\mathcal{R}_1^d}\left( \prod_{i=1}^{n} \mathcal{X}_i, \prod_{i=1}^{n}(M_i + \varepsilon_i) - \prod_{i=1}^{n} M_i \right) \leq \prod_{i=1}^{n} M_{\mathcal{R}_1^d}(\mathcal{X}_i, \varepsilon_i)$$

6. *If for any maximal $R \in \mathcal{S}$ (w.r.t. PSD order), $Tr(R) \geq 1$. Fix $\mathcal{X} \subset B_M^{d_1}$, and $\mathcal{L} \subset \mathbb{R}^{d_2, d_1}$ matrices with spectral norm $\leq r$. Denote $\|A\|_{\mathcal{S}} := \min\{t > 0 : \frac{1}{t} A \in \mathcal{S}\}$, then*

$$M_{\mathcal{S}}\left( \mathcal{L}\mathcal{X}, \varepsilon_2 \sqrt{2r^2 + 2\varepsilon_1^2 \|I_{d_1}\|_{\mathcal{S}}} + \varepsilon_1 M \right) \leq M_{\mathcal{R}_1^d}(\mathcal{L}, \varepsilon_1) \cdot M_{\mathcal{S}}(\mathcal{X}, \varepsilon_2)$$

We next prove each item separately.

*Proof.* (of item 1.) Let $\check{\mathcal{X}}$ be an $\varepsilon$-multicover of $\mathcal{X}$ w.r.t. $\mathcal{S}$. It is enough to show that $A\check{\mathcal{X}}$ is an $(\|A\|\varepsilon)$-multicover of $A\mathcal{X}$. Fix $\mathcal{X} \in \mathcal{X}$ and a PSD matrix $R \in \mathcal{S}$. We need to show that there is $\check{\mathbf{x}} \in \check{\mathcal{X}}$ such that $\|A\mathbf{x} - A\check{\mathbf{x}}\|_R^2 \leq \mathrm{Tr}(R)\|A\|^2\varepsilon^2$. Now, for any $\check{\mathbf{x}}$ we have

$$\|A\mathbf{x} - A\check{\mathbf{x}}\|_R^2 = \|A\|^2(\mathbf{x} - \check{\mathbf{x}})^\top \frac{A^\top}{\|A\|} R \frac{A}{\|A\|}(\mathbf{x} - \check{\mathbf{x}}) = \|A\|^2\|\mathbf{x} - \check{\mathbf{x}}\|_{\frac{A^\top}{\|A\|} R \frac{A}{\|A\|}}^2$$

Finally, since $\check{\mathcal{X}}$ is an $\varepsilon$-multicover w.r.t. $\mathcal{S}$, there is $\check{\mathbf{x}} \in \check{\mathcal{X}}$ such that $\|\mathbf{x} - \check{\mathbf{x}}\|_{\frac{A^\top}{\|A\|} R \frac{A}{\|A\|}}^2 \leq \mathrm{Tr}(\frac{1}{\|A\|} A^\top R \frac{1}{\|A\|} A)\varepsilon^2 \leq \mathrm{Tr}(R)\varepsilon^2$. Therefore overall

$$\|A\mathbf{x} - A\check{\mathbf{x}}\|_R^2 = \|A\|^2\|\mathbf{x} - \check{\mathbf{x}}\|_{\frac{A^\top}{\|A\|} R \frac{A}{\|A\|}}^2 \leq \mathrm{Tr}(R)\|A\|^2\varepsilon^2$$

$\square$

*Proof.* (of item 5.) We first prove the item for $n = 2$. We will then show that the general case follows by induction. In the proof of this item we will denote by $A \circ B$ the elementwise product of two $d \times d$ matrices, and by $\mathrm{diag}(A)$ the diagonal matrix obtained by zeroing the non-diagonal entries of $A$.

Let $\check{\mathcal{X}}_i$ be an $\varepsilon_i$-multicover of $\mathcal{X}_i$ w.r.t. $\mathcal{R}_1^d$. Fix $\mathbf{x}_i \in \mathcal{X}_i$ and a PSD matrix $R \geq 0$ with $\mathrm{Tr}(R) \leq 1$. It is enough to show that there is $\check{\mathbf{x}}_i \in \check{\mathcal{X}}_i$ with $\|\mathbf{x}_1\mathbf{x}_2 - \check{\mathbf{x}}_1\check{\mathbf{x}}_2\|_R \leq M_1\varepsilon_2 + M_2\varepsilon_1 + \varepsilon_1\varepsilon_2$. We have

$$\begin{aligned}\|\mathbf{x}_1\mathbf{x}_2 - \check{\mathbf{x}}_1\check{\mathbf{x}}_2\|_R &\leq \|\mathbf{x}_1\mathbf{x}_2 - \mathbf{x}_1\check{\mathbf{x}}_2\|_R + \|\mathbf{x}_1\check{\mathbf{x}}_2 - \check{\mathbf{x}}_1\check{\mathbf{x}}_2\|_R \\ &= \|\mathbf{x}_2 - \check{\mathbf{x}}_2\|_{R\circ\mathbf{x}_1\mathbf{x}_1^\top} + \|\mathbf{x}_1 - \check{\mathbf{x}}_1\|_{R\circ\check{\mathbf{x}}_2\check{\mathbf{x}}_2^\top}\end{aligned}$$

Now, $\mathrm{Tr}(R \circ \check{\mathbf{x}}_2\check{\mathbf{x}}_2^\top) = \|\check{\mathbf{x}}_2\|_{\mathrm{diag}(R)}^2$. Thus, we can choose $\check{\mathbf{x}}_1$ such that $\|\mathbf{x}_1 - \check{\mathbf{x}}_1\|_{R\circ\check{\mathbf{x}}_2\check{\mathbf{x}}_2^\top} \leq \|\check{\mathbf{x}}_2\|_{\mathrm{diag}(R)}\varepsilon_1$. We get for any $0 < p < 1$

$$\|\mathbf{x}_1\mathbf{x}_2 - \check{\mathbf{x}}_1\check{\mathbf{x}}_2\|_R \overset{(*)}{\leq} \|\mathbf{x}_2 - \check{\mathbf{x}}_2\|_{\frac{1}{p}R\circ\mathbf{x}_1\mathbf{x}_1^\top + \frac{1}{1-p}\mathrm{diag}(\varepsilon_1^2 R)} + M_2\varepsilon_1$$

Where $(*)$ follows from straight-forward calculations that appear fully in the appendix version of this proof. Now, we can choose $\check{\mathbf{x}}_2 \in \check{\mathcal{X}}_2$ with

$$\|\mathbf{x}_2 - \check{\mathbf{x}}_2\|_{\frac{1}{p}R\circ\mathbf{x}_1\mathbf{x}_1^\top + \frac{1}{1-p}\mathrm{diag}(\varepsilon_1^2 R)} \leq \varepsilon_2\sqrt{\mathrm{Tr}\left(\frac{1}{p}R \circ \mathbf{x}_1\mathbf{x}_1^\top + \frac{1}{1-p}\mathrm{diag}(\varepsilon_1^2 R)\right)}$$

$$\leq \varepsilon_2\sqrt{M_1^2/p + \varepsilon_1^2/(1-p)}$$

for $p = M_1/(M_1 + \varepsilon_1)$ we get that

$$\|\mathbf{x}_1\mathbf{x}_2 - \check{\mathbf{x}}_1\check{\mathbf{x}}_2\|_R \leq \varepsilon_2(M_1 + \varepsilon_1) + M_2\varepsilon_1 = M_1\varepsilon_2 + M_2\varepsilon_1 + \varepsilon_1\varepsilon_2$$

We next consider $n > 2$ and conclude the proof by induction. Denote $\mathcal{X}_2' = \prod_{i=2}^n \mathcal{X}_i$, $M_2' = \prod_{i=2}^n M_i$ and $\varepsilon_2' = \prod_{i=2}^n(M_i + \varepsilon_i) - \prod_{i=2}^n M_i$.

By the induction hypothesis we have $M(\mathcal{X}_2', \varepsilon_2') \leq \prod_{i=2}^n M(\mathcal{X}_i, \varepsilon_i)$. By the case $n = 2$ we have

$$M(\mathcal{X}_1\mathcal{X}_2', M_1\varepsilon_2' + M_2'\varepsilon_1 + \varepsilon_1\varepsilon_2') \leq M(\mathcal{X}_1, \varepsilon_1) M(\mathcal{X}_2', \varepsilon_2')$$

$$\leq M(\mathcal{X}_1, \varepsilon_1)\prod_{i=2}^n M(\mathcal{X}_i, \varepsilon_i) = \prod_{i=2}^n M(\mathcal{X}_i, \varepsilon_i)$$

this concludes the proof as $\mathcal{X}_1\mathcal{X}_2' = \prod_{i=1}^n \mathcal{X}_i$ and

$$\begin{aligned}M_1\varepsilon_2' + M_2'\varepsilon_1 + \varepsilon_1\varepsilon_2' &= (M_1 + \varepsilon_1)\left(\prod_{i=2}^n(M_i + \varepsilon_i) - \prod_{i=2}^n M_i\right) + \varepsilon_1\prod_{i=2}^n M_i \\ &= \prod_{i=1}^n(M_i + \varepsilon_i) - \prod_{i=1}^n M_i\end{aligned}$$

$\square$

*Proof.* (of item 6) For a nice set $\mathcal{S} \subset \mathcal{R}^d$, and PSD matrix $R \in \mathbb{R}^{d \times d}$, define $\|R\|_{\mathcal{S}} = \min\{t > 0 : \frac{1}{t}R \in \mathcal{S}\}$. Note that this is almost a norm - the triangle inequality, positive definiteness, and homogeneity for positive scalars apply - but do not apply for negative scalars. Let $\check{\mathcal{L}}$ be an $\varepsilon_1$-multicover of $\mathcal{L}$ w.r.t. $\mathcal{R}_1^d$ and let $\check{\mathcal{X}}$ be an $\varepsilon_2$-multicover of $\mathcal{X}$ w.r.t. $\mathcal{S}$. We will show that $\check{\mathcal{L}}\check{\mathcal{X}}$ is an $\varepsilon_2\sqrt{2r^2 + 2\varepsilon_1\|I_{d_1}\|_{\mathcal{S}}} + \varepsilon_1 M$-multicover of $\mathcal{LX}$ w.r.t. $\mathcal{S}$. Fix $R \in \mathcal{S}$. W.l.o.g we may assume that it is maximal w.r.t. PSD order. Let $W \in \mathcal{L}$ and $\mathbf{x} \in \mathcal{X}$. We need to show that there are $\check{W} \in \check{\mathcal{L}}$ and $\check{\mathbf{x}} \in \check{\mathcal{X}}$ with $\|W\mathbf{x} - \check{W}\check{\mathbf{x}}\|_R \leq \varepsilon_2\sqrt{2\mathrm{Tr}(R)r^2 + 2\varepsilon_1\mathrm{Tr}(R)\|I_{d_1}\|_{\mathcal{S}}} + \varepsilon_1\sqrt{\mathrm{Tr}(R)}M$. We have

$$\|W\mathbf{x} - \check{W}\check{\mathbf{x}}\|_R \leq \|W\mathbf{x} - W\check{\mathbf{x}}\|_R + \|W\check{\mathbf{x}} - \check{W}\check{\mathbf{x}}\|_R = \|\mathbf{x} - \check{\mathbf{x}}\|_{W^\top RW} + \|(W - \check{W})\check{\mathbf{x}}\|_R$$

$$= \|\mathbf{x} - \check{\mathbf{x}}\|_{W^\top RW} + \sqrt{\check{\mathbf{x}}^\top(W - \check{W})^\top R(W - \check{W})\check{\mathbf{x}}}$$

Now, $(W_1, W_2) \mapsto \check{\mathbf{x}}^\top W_1^\top RW_2\check{\mathbf{x}}$ is a symmetric and positive bi-linear form on the space of $d_2 \times d_1$ matrices of trace

$$\sum_{i=1}^{d_2}\sum_{j=1}^{d_1} \check{\mathbf{x}}^\top E_{ij}^\top RE_{ij}\check{\mathbf{x}} = \sum_{i=1}^{d_2}\sum_{j=1}^{d_1} (\check{x}_j\mathbf{e}_i)^\top R(\check{x}_j\mathbf{e}_i) = \sum_{i=1}^{d_2}\sum_{j=1}^{d_1} \check{x}_j^2 R_{ii} = \mathrm{Tr}(R)\|\check{\mathbf{x}}\|^2$$

Thus, there is $\check{W} \in \check{\mathcal{L}}$ such that $\check{\mathbf{x}}^\top(W - \check{W})^\top R(W - \check{W})\check{\mathbf{x}} \leq \mathrm{Tr}(R)\|\check{\mathbf{x}}\|^2\varepsilon_1^2$. For this $\check{W}$ we have

$$\|W\mathbf{x} - \check{W}\check{\mathbf{x}}\|_R \overset{(*)}{\leq} \sqrt{2}\|\mathbf{x} - \check{\mathbf{x}}\|_{W^\top RW + \varepsilon_1^2\mathrm{Tr}(R)I_{d_1}} + \varepsilon_1\sqrt{\mathrm{Tr}(R)}M$$

Where $(*)$ follows from simple calculations, that appear fully in the appendix version of this proof. Thus, it is possible to choose $\check{\mathbf{x}} \in \check{\mathcal{X}}$ s.t. $\|\mathbf{x} - \check{\mathbf{x}}\|_{W^\top RW + \varepsilon_1^2 I_{d_1}} \leq \varepsilon_2\sqrt{\|W^\top RW + \varepsilon_1^2\mathrm{Tr}(R)I_{d_1}\|_{\mathcal{S}}}$. Finally, $\|W^\top RW + \varepsilon_1^2\mathrm{Tr}(R)I_{d_1}\|_{\mathcal{S}} \leq r^2 + \varepsilon_1^2\mathrm{Tr}(R)\|I_{d_1}\|_{\mathcal{S}} \leq Tr(R)r^2 + \varepsilon_1^2\mathrm{Tr}(R)\|I_{d_1}\|_{\mathcal{S}}$ $\square$

**Lemma 2.7.** *Fix* $\mathcal{X} \subset \mathbb{R}^d$, $\varepsilon > 0$ *and* $r > 2$. *It holds that* $N_\infty(\mathcal{X}, r\varepsilon) \leq \left(M_{\mathcal{R}_1^d}(\mathcal{X}, \varepsilon)\right)^{\lceil \log_{r/2}(d) \rceil}$.

*Proof.* Let $\check{\mathcal{X}}$ be an $\varepsilon$-multicover of $\mathcal{X}$ of size $M(\mathcal{X}, \varepsilon)$. By lemma 2.4 for any $\mathbf{x} \in \mathcal{X}$ there is a distribution $\mathcal{D}_\mathbf{x}$ on $\check{\mathcal{X}}$ such that if $X \sim \mathcal{D}_\mathbf{x}$ then $X$ is an $\varepsilon$-estimator of $\mathbf{x}$. In particular, for any coordinate $i \in [d]$ we have

$$\mathbb{E}_X(X_i - x_i)^2 = \mathbb{E}_X(X - \mathbf{x})^\top E_{ii}(X - \mathbf{x}) \leq \varepsilon^2$$

Denote $k = \lceil \log_{r/2}(d) \rceil$. By the above equation and lemma 2.1 we conclude that if $X^1, \ldots, X^k \sim \mathcal{D}_\mathbf{x}$ then for every $i \in [d]$

$$\Pr\left(\exists i \in [d] \text{ s.t. } |\mathrm{median}(X_i^1, \ldots, X_i^k) - x_i| > r\varepsilon\right) < d\left(\frac{2}{r}\right)^k \leq 1$$

in particular, there exists $\mathbf{x}^1, \ldots, \mathbf{x}^k \in \check{\mathcal{X}}$ such that for any $i \in [d]$, $|\mathrm{median}(x_i^1, \ldots, x_i^k) - x_i| \leq r\varepsilon$. This implies that

$$\mathrm{median}(\check{\mathcal{X}}^k) := \left\{\left(\mathrm{median}(x_1^1, \ldots, x_1^k), \ldots, \mathrm{median}(x_d^1, \ldots, x_d^k)\right) : \mathbf{x}^1, \ldots, \mathbf{x}^k \in \check{\mathcal{X}}\right\}$$

is an $\varepsilon$-cover of $\mathcal{X}$ w.r.t. the $\ell^\infty$ norm. This concludes the proof as $|\mathrm{median}(\check{\mathcal{X}}^k)| \leq |\check{\mathcal{X}}|^k$ $\square$

## 3 MULTICOVER FOR NEURAL NETWORKS SAMPLE COMPLEXITY

### 3.1 MULTICOVER FOR SEQUENCE OF VECTORS

We denote by $\mathbb{R}^{d,m}$ the vector space of sequences $\mathbf{x} = (\mathbf{x}^1, \ldots, \mathbf{x}^m)$ of $m$ vectors in $\mathbb{R}^d$. We next extend the notion of multicover, as well as multicover calculus, to subsets $\mathbb{R}^{d,m}$. This extension is useful for sample complexity analysis via multicover.

Denote by $\mathbb{S}^{d,m}$ the collection of all sequences $(\mathbf{u}_1, \ldots, \mathbf{u}_m) \in \mathbb{R}^{d,m}$ with $\sum_{i=1}^m \|\mathbf{u}_i\|^2 = 1$. We also denote by $\mathcal{R}^{d,m}$ the convex set of sequences $R = (R_1, \ldots, R_m)$ of $m$ $d \times d$ PSD matrices.

We denote $\mathrm{Tr}(R) = \sum_{i=1}^m \mathrm{Tr}(R_i)$ and $\mathcal{R}_t^{d,m} = \{R \in \mathcal{R}^{d,m} : \mathrm{Tr}(R) \leq t\}$. We say that a set $\mathcal{S} \subset \mathcal{R}^{d,m}$ is *nice*, if $\forall R \in \mathcal{S}$, $i \in [m]$ and $W \in \mathbb{R}^{d \times d,m}$ with $\|W_i\| \leq 1$ then $W^\top RW = (W_1^\top R_1 W_1, \ldots, W_m^\top R_m W_m) \in \mathcal{S}$. We denote the inner product, norm and metric induced by $R \in \mathcal{R}^{d,m}$ on $\mathbb{R}^{d,m}$ by $\langle \mathbf{x}, \mathbf{y} \rangle_R = \sum_{i=1}^m \langle \mathbf{x}^i, R_i \mathbf{y}^i \rangle$, $\|\mathbf{x}\|_R = \sqrt{\langle \mathbf{x}, \mathbf{x} \rangle_R}$ and $d_R(\mathbf{x}, \mathbf{y}) = \|\mathbf{x} - \mathbf{y}\|_R$. A set $\check{\mathcal{X}} \subset \mathbb{R}^{d,m}$ is an *$\varepsilon$-multicover* of $\mathcal{X}$ w.r.t. a nice set $\mathcal{S}$ if for any $R \in \mathcal{S}$ and every $\mathbf{x} \in \mathcal{X}$ there is $\check{\mathbf{x}} \in \check{\mathcal{X}}$ such that $\|\mathbf{x} - \check{\mathbf{x}}\|_R \leq \sqrt{\mathrm{Tr}(R)}\varepsilon$. Equivalently, for any $R \in \mathcal{R}_1^{d,m}$ $\check{\mathcal{X}}$ is an $\varepsilon$-cover of $\mathcal{X}$ w.r.t. $d_R$. The *$\varepsilon$-multicovering-number* of $\mathcal{X}$, denoted by $M(\mathcal{X}, \varepsilon)$ is the minimal size of an $\varepsilon$-multicover of $\mathcal{X}$.

We will use $N_R(\mathcal{X}, \varepsilon)$ for the covering number w.r.t. the metric $d_R$, $N_\infty(\mathcal{X}, \varepsilon)$ when the metric is $d(\mathbf{x}, \mathbf{y}) = \max_{j \in [m]} \|\mathbf{x}^j - \mathbf{y}^j\|_\infty$ and $N_2(\mathcal{X}, \varepsilon)$ when the metric is $d(\mathbf{x}, \mathbf{y}) = \sqrt{\frac{1}{m} \sum_{j=1}^m \|\mathbf{x}^j - \mathbf{y}^j\|_2^2}$.

We say that a random variable $X \in \mathbb{R}^{d,m}$ is an *$\varepsilon$-estimator* of $\mathbf{x} \in \mathbb{R}^{d,m}$ if for any $\mathbf{u} \in \mathbb{S}^{d,m}$, we have $\sum_{j=1}^m \mathbb{E}\langle \mathbf{u}^j, X^j - \mathbf{x}^j \rangle^2 \leq \varepsilon^2$. Equivalently, for any $R \in \mathcal{R}^{d,m}$, $\mathbb{E}\|X - \mathbf{x}\|_R^2 \leq \mathrm{Tr}(R)\varepsilon^2$.

We next generalize Lemmas 2.4, 2.6 and 2.7 to the extended definition of multicover. The proofs of the generalized lemmas are similar to the proofs of the original lemmas and are deffered to the appendix, similarly to the other absent proofs.

**Lemma 3.1.** *Let $\mathcal{X} \subset \mathbb{R}^{d,m}$. A set $\check{\mathcal{X}} \subset \mathbb{R}^{d,m}$ is an $\varepsilon$-multicover of $\mathcal{X}$ w.r.t. $\mathcal{R}^{d,m}$ if and only if for any $\mathbf{x} \in \mathcal{X}$ there is a random vector $X \in \check{\mathcal{X}}$ that is an $\varepsilon$-estimator of $\mathbf{x}$.*

**Lemma 3.2.**   *1. For $\mathcal{X} \subset \mathbb{R}^{d_1,m}$ and $A \in \mathbb{R}^{d_2 \times d_1}$ we have $M_{\mathcal{S}}(A\mathcal{X}, \|A\|\varepsilon) \leq M_{\mathcal{S}}(\mathcal{X}, \varepsilon)$*

2. *For $\mathcal{X}_1, \ldots, \mathcal{X}_n \subset \mathbb{R}^{d,m}$ and $\varepsilon_1, \ldots, \varepsilon_n > 0$ we have $M_{\mathcal{S}}(\sum_{i=1}^n \mathcal{X}_i, \sum_{i=1}^n \varepsilon_i) \leq \prod_{i=1}^n M_{\mathcal{S}}(\mathcal{X}_i, \varepsilon_i)$*

3. *For $\mathcal{X} \subset \mathbb{R}^{d,m}, \varepsilon > 0$ and $\mathbf{b} \in \mathbb{R}^{d,m}$ we have $M_{\mathcal{S}}(U\mathcal{X} + \mathbf{b}, \varepsilon) = M_{\mathcal{S}}(\mathcal{X}, \varepsilon)$*

4. *For $\mathcal{X}_1, \ldots, \mathcal{X}_n \subset \mathbb{R}^{d,m}$ and $\varepsilon > 0$ we have $M_{\mathcal{S}}(\cup_{i=1}^n \mathcal{X}_i, \varepsilon) \leq \sum_{i=1}^n M_{\mathcal{S}}(\mathcal{X}_i, \varepsilon)$*

5. *For $\mathcal{S} = \mathcal{R}_1^{d,m}$ $\mathcal{X}_i \subset [-M_i, M_i]^{d,m}$ and $\varepsilon_1, \ldots, \varepsilon_n > 0$ we have*

$$M_{\mathcal{R}_1^{d,m}}\left(\prod_{i=1}^n \mathcal{X}_i, \prod_{i=1}^n (M_i + \varepsilon_i) - \prod_{i=1}^n M_i\right) \leq \prod_{i=1}^n M_{\mathcal{R}_1^{d,m}}(\mathcal{X}_i, \varepsilon_i)$$

6. *For[1] $\mathcal{S} = \mathcal{R}_1^{d,m}$, fix $\mathcal{X} \subset B_M^{d_1,m}$, $\mathcal{L} \subset \mathbb{R}^{d_2,d_1}$ matrices with spectral norm $\leq r$. Then,*

$$M_{\mathcal{S}}\left(\mathcal{L}\mathcal{X}, \varepsilon_2\sqrt{2r^2 + 2\varepsilon_1^2 d_1} + \varepsilon_1 M\right) \leq M_{\mathcal{R}_1^{d,m}}(\mathcal{L}, \varepsilon_1) \cdot M_{\mathcal{S}}(\mathcal{X}, \varepsilon_2)$$

**Lemma 3.3.** *Fix $\mathcal{X} \subset \mathbb{R}^{d,m}$, $\varepsilon > 0$ and $r > 2$.*

*Then $N_\infty(\mathcal{X}, r\varepsilon) \leq \left(M_{\mathcal{R}_1^{d,m}}(\mathcal{X}, \varepsilon)\right)^{\lceil \log_{r/2}(dm) \rceil}$.*

### 3.1.1 STRONGLY BOUNDED ACTIVATION

In this section we will develop tools to calculate $M_{\mathcal{R}_1^{d,m}}(\rho(\mathcal{X}), \varepsilon)$ for a smooth enough $\rho$. For the sake of cleanliness we will denote $M(\cdot, \cdot) := M_{\mathcal{R}_1^{d,m}}(\cdot, \cdot)$. The smoothness requirements are given in the following definition.

**Definition 3.4.** *A function $\rho : \mathbb{R} \to \mathbb{R}$ is $B$-strongly-bounded if for all $n \geq 1$, $\|\rho^{(n)}\|_\infty \leq n!B^n$. Likewise, $\rho$ is strongly-bounded if it is $B$-strongly-bounded for some $B$*

As shown in Daniely & Granot (2019) the ReLU-like function $\log(1 + e^x)$ is strongly bounded, as well as the sigmoid function $\frac{e^x}{1+e^x}$. It is also shown in Daniely & Granot (2019) that

---

[1]This claim can be generalized to a more general $\mathcal{S}$. We present the case $\mathcal{S} = \mathcal{R}_1^{d,m}$ for simplicity.

**Fact 3.5.** *If $\rho$ is $B$-strongly-bounded then $\rho$ is analytic and its Taylor coefficients around any point are bounded by $B^n$ for any $n \geq 1$.*

We will utilize this fact in order to calculate the effect of a non-linearity on the multicovering number.

**Lemma 3.6** ($\beta$-Swish Activation Ramachandran et al. (2017))**.** *For a constant $\beta \geq 0$, the function $\frac{x}{1+e^{-\beta x}}$ is strongly-bounded*

**Lemma 3.7** (Hyperbolic Tangent)**.** *The function $\frac{e^{2x}-1}{e^{2x}+1}$ is strongly-bounded*

In order to analyze $M(\rho(\mathcal{X}), \varepsilon)$ for a strongly bounded $\rho$, we first analyze $M(p(\mathcal{X}), \varepsilon)$ for a polynomial $p$, and then utilize fact 3.5.

**Lemma 3.8.** *Let $p(x) = \sum_{i=0}^{k} a_i X^i$ be a polynomial with $|a_i| \leq B^i$ and suppose that $\mathcal{X} \subset \left[-\frac{1}{8B}, \frac{1}{8B}\right]^{d,m}$. Then, for any Let $0 < \varepsilon \leq 1$, $M\left(p(\mathcal{X}), \varepsilon\right) \leq \left(M\left(\mathcal{X}, \frac{\varepsilon}{8B}\right)\right)^{\frac{k(k+1)}{2}}$*

We are now ready to present our main tool for analysing $M(\rho(\mathcal{X}), \varepsilon)$ for strongly bounded $\rho$.

**Lemma 3.9.** *Let $\mathcal{X} \subset \mathbb{R}^{d,m}$. Let $\rho : \mathbb{R} \to \mathbb{R}$ be $B$ strongly bounded. Then for $1 \geq \varepsilon > 0$,*

$$M(\rho(\mathcal{X}), \varepsilon + \sqrt{d}8^{-(k+1)}) \leq \left(M\left(\mathcal{X}, \frac{1}{32B}\right)\right)^{\lceil \log_2(dm) \rceil} \left(M\left(\mathcal{X}, \frac{\varepsilon}{8B}\right)\right)^{\frac{k(k+1)}{2}}$$

*Proof.* Assume first that $\mathcal{X} \subset \mathbb{R}^{d,m}$ is contained in an $\ell^\infty$ ball of radius $\frac{1}{8B}$. Since multicovering numbers are invariant to translations (i.e. $M(\mathcal{X}, \varepsilon) = M(\mathcal{X} + \mathbf{b}, \varepsilon)$ for any $\mathbf{b} \in \mathbb{R}^{m,d}$), we can assume w.l.o.g. that $\mathcal{X} \subset \left[-\frac{1}{8B}, \frac{1}{8B}\right]^{d,m}$. Let $p$ be the Taylor polynomial of $\rho$ around $0$ of degree $k$ and let $r = \rho - p$. We have that for any $x \in \left[-\frac{1}{8B}, \frac{1}{8B}\right]$, $|r(x)| \leq B^{k+1}|x|^{k+1} \leq 8^{-(k+1)}$. Thus $\{0\}$ is an $\left(\sqrt{d}8^{-(k+1)}\right)$-multicover of $r(\mathcal{X})$. Indeed, if $R \in \mathcal{R}_1^{d,m}$ and $\mathbf{x} \in r(\mathcal{X})$ then

$$\|\mathbf{x}\|_R^2 = \sum_{i=1}^{m} \left\langle \mathbf{x}^i, R_i \mathbf{x}^i \right\rangle \leq \sum_{i=1}^{m} \text{Tr}(R_i)\|\mathbf{x}^i\|^2$$

$$\leq \sum_{i=1}^{m} \text{Tr}(R_i)d8^{-2(k+1)} = d8^{-2(k+1)}\text{Tr}(R) \leq d8^{-2(k+1)}$$

In particular $M(r(\mathcal{X}), \sqrt{d}8^{-(k+1)}) = 1$. Now, we have

$$M(\rho(\mathcal{X}), \varepsilon + \sqrt{d}8^{-(k+1)}) \overset{\overset{\rho(\mathcal{X}) \subset p(\mathcal{X})+r(\mathcal{X})}{\leq}}{} M(p(\mathcal{X}) + r(\mathcal{X}), \varepsilon + \sqrt{d}8^{-(k+1)})$$

$$\overset{\overset{\text{Lemma 3.2}}{\leq}}{} M\left(r(\mathcal{X}), \sqrt{d}8^{-(k+1)}\right) M(p(\mathcal{X}), \varepsilon)$$

$$\overset{\overset{\text{Lemma 3.8}}{\leq}}{} \left(M\left(\mathcal{X}, \frac{\varepsilon}{8B}\right)\right)^{\frac{k(k+1)}{2}}$$

Finally, by lemma 3.3, $\mathcal{X}$ is a union of $\left(M\left(\mathcal{X}, \frac{1}{32B}\right)\right)^{\lceil \log_2(dm) \rceil}$ sets $\mathcal{X}_i$, such that each $\mathcal{X}_i$ is contained in an $\ell^\infty$ ball of radius $\frac{1}{8B}$. Applying the above argument to each $\mathcal{X}_i$ implies the lemma. $\qquad\square$

### 3.2 Neural Network Sample Complexity via Multicover

#### 3.2.1 Multicover and sample complexity

Fix an instance space $\mathcal{Z}$, a label space $\mathcal{Y}$ and a loss $\ell : \mathbb{R}^d \times \mathcal{Y} \to [0, \infty)$. We say that $\ell$ has some property $p$ (e.g. boundness, Lipschitzness, etc.) if for any $y \in \mathcal{Y}$, $\ell(\cdot, y)$ has the property $p$. Fix a class $\mathcal{H}$ from $\mathcal{Z}$ to $\mathbb{R}^d$. For a distribution $\mathcal{D}$ and a sample $S \in (\mathcal{Z} \times \mathcal{Y})^m$ we define the *representativeness* of $S$ as

$$\text{rep}_{\mathcal{D}}(S, \mathcal{H}) = \sup_{h \in \mathcal{H}} \ell_{\mathcal{D}}(h) - \ell_S(h)$$

Where $\ell_{\mathcal{D}}(h) = \mathbb{E}_{(x,y) \sim \mathcal{D}} \ell(h(x), y)$ and $\ell_S(h) = \frac{1}{m} \sum_{i=1}^{m} \ell(h(x_i), y_i)$. We note that if $\text{rep}_{\mathcal{D}}(S, \mathcal{H}) \leq \varepsilon$ then any algorithm that is guaranteed to return a function $\hat{h} \in \mathcal{H}$ will enjoy a

generalization bound $\ell_{\mathcal{D}}(h) \leq \ell_S(h) + \varepsilon$. In particular, the ERM algorithm will return a function whose loss is optimal, up to an additive factor of $\varepsilon$.

We will focus on bounds on $\text{rep}_{\mathcal{D}}(S, \mathcal{H})$ when $S \sim \mathcal{D}^m$. To this end, we will rely on the connection between representativeness and the *covering numbers* of $\mathcal{H}$. For $x_1, \ldots, x_m \in \mathcal{Z}$ we denote

$$\mathcal{H}(x_1, \ldots, x_m) = \{(h(x_1), \ldots, h(x_m)) : h \in \mathcal{H}\}$$

Given a metric $d$ on $\mathbb{R}^{d,m}$ we denote $N_d(\mathcal{H}, m, \epsilon) = \sup_{x_1, \ldots, x_m \in \mathcal{Z}} N_d(\mathcal{H}(x_1, \ldots, x_m), m, \varepsilon)$. Similarly, we denote $M(\mathcal{H}, m, \epsilon) = \sup_{x_1, \ldots, x_m \in \mathcal{Z}} M(\mathcal{H}(x_1, \ldots, x_m), m, \varepsilon)$.

**Lemma 3.10.** *(Shalev-Shwartz & Ben-David, 2014) Let $\ell : \mathbb{R}^d \times \mathcal{Y} \to \mathbb{R}$ be $B$-bounded. Then for any distribution $\mathcal{D}$ on $\mathcal{Z}$*

$$\mathbb{E}_{S \sim \mathcal{D}^m} \text{rep}_{\mathcal{D}}(S, \mathcal{H}) \leq B2^{-M+1} + \frac{12B}{\sqrt{m}} \sum_{k=1}^{M} 2^{-k} \sqrt{\ln\left(N_2(\ell \circ \mathcal{H}, m, B2^{-k})\right)}$$

We conclude with a special case of the above lemma, which will be useful in this paper.

**Lemma 3.11.** *Let $\ell : \mathbb{R}^d \times \mathcal{Y} \to \mathbb{R}$ be $L$-Lipschitz w.r.t. $\|\cdot\|_\infty$ and $B$-bounded. Assume that for any $\frac{\sqrt{n}B}{\sqrt{m}8L} \leq \varepsilon \leq 1$, $\ln M(\mathcal{H}, m, \varepsilon) \leq \frac{n}{\varepsilon^2}$. Then for any distribution $\mathcal{D}$ on $\mathcal{Z}$*

$$\mathbb{E}_{S \sim \mathcal{D}^m} \text{rep}_{\mathcal{D}}(S, \mathcal{H}) \lesssim \frac{(L+B)\sqrt{n}}{\sqrt{m}} \sqrt{\log(dm)} \log(m)$$

### 3.2.2 SAMPLE COMPLEXITY OF NEURAL NETWORKS

Fix the instance space to be the ball of radius $\sqrt{d_0}$ in $\mathbb{R}^{d_0}$ (in particular $[-1, 1]^{d_0} \subset \mathcal{X}$). Fix also a $B$-strongly-bounded activation function $\rho$. Consider the class

$$\mathcal{N}_{r,R}^\rho(d_0, \ldots, d_t) = \left\{W_t \circ \rho \circ W_{t-1} \circ \rho \ldots \circ \rho \circ W_1 : W_i \in M_{d_{i-1}d_i} \|W_i\| \leq r, \|W_i\|_F \leq R\right\}$$

and more generally, for matrices $W_i^0 \in M_{d_i, d_{i-1}}$, $i = 1, \ldots, t$ consider

$$\mathcal{H} = \mathcal{N}_{r,R}^\rho(W_1^0, \ldots, W_t^0) = \left\{W_t \circ \rho \circ W_{t-1} \circ \rho \ldots \circ \rho \circ W_1 : \|W_i - W_i^0\| \leq r, \|W_i - W_i^0\|_F \leq R\right\}$$

denote $d = \max(d_0, \ldots, d_t)$. We will assume that $t, \|W_i^0\|, r$ are all bounded by some constant $C > 0$, and will allow hidden constants to depend $C$. This is motivated by the fact that in practice, $\|W_i^0\|, r$ are often bounded by small constants. For instance, if the initial weights are sampled form the standard Xavier initialization then $\|W_i^0\| \approx \sqrt{2}$ w.h.p. for resnets we have $\|W_i^0\| \approx 1$. We will also allow hidden constant to depend on the activation $\rho$ and the depth $t$.

**Theorem 3.12.** *Let $\ell : \mathbb{R}^d \times \mathcal{Y} \to \mathbb{R}$ be $O(1)$-Lipschitz w.r.t. $\|\cdot\|_\infty$ and $O(1)$-bounded. Then for any distribution $\mathcal{D}$ on $\mathcal{Z}$*

$$\mathbb{E}_{S \sim \mathcal{D}^m} \text{rep}_{\mathcal{D}}(S, \mathcal{H}) \lesssim \sqrt{\frac{dR^2}{m}} \log^{t+2}(Rdm)$$

The theorem is implied by the following lemma together with lemma 3.11.

**Lemma 3.13.** *For any $0 < \epsilon \leq 1$, $M(\mathcal{H}, m, \epsilon) \lesssim (\log(dm) + \log^2(d/\epsilon))^t \log(dR) \frac{dR^2}{\epsilon^2}$*

The proof follows a peeling argument, applying lemmas 3.2, 3.9 and 2.5 inductively, for each layer.

## 4 RELATED WORK

In recent years, there has been active work in the area of the sample complexity of neural networks. For the the remaining of this section, we refer the reader to Table 1 to explain the notation used in different works.

Table 1: A table comparing the sample complexity bounds of different recent works. In the "$\tilde{O}(1)$ outputs" column, we adopt the notation of our paper, where $R$, $r$ being the Frobenius and the spectral norm, $t$ being the depth of the network, and $d$ its width. $m$ is the sample size and $\gamma$ is the margin wherever it is used.

| Paper | Result | $\tilde{O}(1)$ outputs | Notation |
|---|---|---|---|
| Golowich et al. (2018) | $\tilde{O}\left(\|\mathbf{x}\|_2 \left(\prod_{j=1}^{t} \|W_j\|_F\right) \cdot \min\left\{\sqrt{\frac{\log\left(\frac{1}{\Gamma}\prod_{j=1}^{t}\|W_j\|_F\right)}{\sqrt{m}}}, \sqrt{\frac{t}{m}}\right\}\right)$ | $\tilde{O}\left(\sqrt{d}R^t \cdot \min\left\{\sqrt{\frac{\log\left(\frac{1}{\Gamma}R^t\right)}{\sqrt{m}}}, \sqrt{\frac{t}{m}}\right\}\right)$ | $\Gamma$ lower bound on product of spectral norms. |
| Bartlett et al. (2017) | $\hat{\mathcal{R}}_\gamma(NN_W) + \tilde{O}\left(\frac{\|\mathbf{x}\|_2\left(\prod_{i=1}^{t}\rho_i\|W_i\|_2\right)\left(\sum_{i=1}^{t}\frac{\|W_i^\top - W_i^{0\top}\|_{2,1}^{2/3}}{\|W_i\|_2^{2/3}}\right)^{3/2}}{\gamma\sqrt{m}}\ln(d) + \sqrt{\frac{\ln(1/\delta)}{m}}\right)$ | $\hat{\mathcal{R}}_\gamma(NN_W) + \tilde{O}\left(\frac{r^t\sqrt{d}\left(\sum_{i=1}^{t}\frac{\|W_i^\top - W_i^{0\top}\|_{2,1}^{2/3}}{r^{2/3}}\right)^{3/2}}{\gamma\sqrt{m}}\ln(d) + \sqrt{\frac{\ln(1/\delta)}{m}}\right)$ | $\hat{\mathcal{R}}_\gamma(f) \leq m^{-1}\sum_i \mathbb{1}\left[f(x_i)_{y_i} \leq \gamma + \max_{j\neq y_i} f(x_i)_j\right]$ is the empirical margin loss |
| Hsu et al. (2021) | $\tilde{O}\left(\frac{\|X\|_F}{m^{3/4}}\left[\prod_j\|W\|_2\right]\left[\sum_i\left(\frac{\|W_i\|_F}{\|W_i\|_2}\right)^{4/5}\right]^{5/4}\left[\sum_i\ln\|W_i\|_F\right]^{1/4}\right)$ | $\tilde{O}\left(\frac{\sqrt{d}}{m^{1/4}}r^t\left[\sum_i\left(\frac{R}{r}\right)^{4/5}\right]^{5/4}\left[\sum_i\ln R\right]^{1/4}\right)$ | |
| Neyshabur et al. (2015) | $\sqrt{\gamma^2\left(2d^{\left[\frac{1}{p^*}-\frac{1}{q}\right]_+}\right)^{2(t-1)}\frac{\min\{p^*,4\log(2D)\}\max_i\|\mathbf{x}_i\|_{p^*}^2}{m}}$ | $\sqrt{R^2t\frac{2d}{m}}$ | $D$ is the input size. $1 \leq q$, $1 \leq p < \infty$, $\gamma = \prod\|W_i\|_{p,q}$ |
| Neyshabur et al. (2017) | $\hat{L}_\gamma(f_w) + \mathcal{O}\left(\sqrt{\frac{\|\mathbf{x}\|_2^2 t^2 d\ln(td)\prod_{i=1}^{t}\|W_i\|_2^2\sum_{i=1}^{t}\frac{\|W_i\|_F^2}{\|W_i\|_2^2}+\ln\frac{tm}{\delta}}{\gamma^2 m}}\right)$ | $\hat{L}_\gamma(f_w) + \mathcal{O}\left(\sqrt{\frac{t^3 r^t d^2\ln(td)\frac{R^2}{r^2}+\ln\frac{tm}{\delta}}{\gamma^2 m}}\right)$ | $\hat{L}_\gamma(f) \leq m^{-1}\sum_i\mathbb{1}\left[f(x_i)_{y_i} \leq \gamma + \max_{j\neq y_i}f(x_i)_j\right]$ is the empirical margin loss |
| Ours[2] | $\tilde{O}\left(\frac{\sqrt{d}RB^t r^t}{\sqrt{m}}\right)$ | $\tilde{O}\left(\frac{\sqrt{d}Rr^t}{\sqrt{m}}\right)$ | Where $B$ is the strongly-boundedness constant |

Neyshabur et al. (2015) give a bound of $\sqrt{\gamma^2\left(2d^{\left[\frac{1}{p^*}-\frac{1}{q}\right]_+}\right)^{2(t-1)}\frac{\min\{p^*,4\log(2D)\}\max_i\|\mathbf{x}_i\|_{p^*}^2}{m}}$ using a peeling argument on the Rademacher complexity of neural networks. Bartlett et al. (2017) use a peeling argument on the covering number of neural networks and yield a bound of $\hat{\mathcal{R}}_\gamma(NN_W) +$

$$\tilde{O}\left(\frac{\|\mathbf{x}\|_2\left(\prod_{i=1}^{t}\rho_i\|W_i\|_2\right)\left(\sum_{i=1}^{t}\frac{\|W_i^\top - W_i^{0\top}\|_{2,1}^{2/3}}{\|W_i\|_2^{2/3}}\right)^{3/2}}{\gamma\sqrt{m}}\ln(d) + \sqrt{\frac{\ln(1/\delta)}{m}}\right).$$ Neyshabur et al. (2017) use a Pac-

Bayes argument to produce a bound of $\hat{L}_\gamma(f_w) + \mathcal{O}\left(\sqrt{\frac{\|\mathbf{x}\|_2^2 t^2 d\ln(td)\prod_{i=1}^{t}\|W_i\|_2^2\sum_{i=1}^{t}\frac{\|W_i\|_F^2}{\|W_i\|_2^2}+\ln\frac{tm}{\delta}}{\gamma^2 m}}\right).$

Golowich et al. (2018) use a Jensen inequality trick to enhance the peeling argument of Neyshabur et al. (2015) and yield $\tilde{O}\left(\|\mathbf{x}\|_2\left(\prod_{j=1}^{t}\|W_j\|_F\right)\cdot\min\left\{\sqrt{\frac{\log\left(\frac{1}{\Gamma}\prod_{j=1}^{t}\|W_j\|_F\right)}{\sqrt{m}}},\sqrt{\frac{t}{m}}\right\}\right).$ Hsu et al. (2021) use the possible existence of a smaller distilled version of a network to obtain the bounds of $\tilde{O}\left(\frac{\|X\|_F}{m^{3/4}}\left[\prod_j\|W\|_2\right]\left[\sum_i\left(\frac{\|W_i\|_F}{\|W_i\|_2}\right)^{4/5}\right]^{5/4}\left[\sum_i\ln\|W_i\|_F\right]^{1/4}\right).$ Vardi et al. (2022) analyze the special case of two-layer neural networks and give a bound of $\tilde{O}\left(\frac{\|\mathbf{w}\|\cdot\|W\|_2\|x\|_2}{\sqrt{m}}\right)$ which is similar to ours when restricting the depth of the network to 2. Finally, Daniely & Granot (2019) provide a bound equivalent to ours, introducing a technique called Approximate Description Length.

We focus on the setting where neural networks have constant depth, and the output of each neuron as well as each input coordinate is $\tilde{O}(1)$. This is usually the case in practice. As shown in table 1, under these setting our work matches the state of the art, and except Daniely & Granot (2019), improves on previous works polynomially. An important caveat to our work, is that it applies only to neural networks with smooth activations. This is similar to Daniely & Granot (2019); Vardi et al. (2022), but the rest of the cited works consider the non-smooth ReLU activations.

The resemblance of the results of Daniely & Granot (2019) to ours is not coincidental. A full discussion of the connection between the notion of Approximate Description Length and multicovering numbers appears in appendix B

## 5 FUTURE DIRECTIONS

Future direction arising from our work Covering-Packing relations for multicover, the (in)Existence of proper multicover (that is, a multicover in which each point is in the class), and the behaviour of multicover w.r.t. Lipschitz functions (specifically, ReLU).

---

[2]Equivalent to Daniely & Granot (2019), and Vardi et al. (2022) result for 2 layer networks

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

# A    OMITTED PROOFS

## A.1    PRELIMINARY LEMMAS

**Lemma A.1** (2.1). *Let $X_1, \ldots, X_k$ be independent r.v. with that that are $\sigma$-estimators to $\mu$. Then*

$$\Pr\left(|\text{median}(X_1, \ldots, X_k) - \mu| > r\sigma\right) < \left(\frac{2}{r}\right)^k$$

*Proof of 2.1.* We have that $\Pr(|X_i - \mu| > r\sigma) \leq \frac{1}{r^2}$. It follows that the probability that $\geq \frac{k}{2}$ of $X_1, \ldots, X_k$ fall outside of the segment $(\mu - r\sigma, \mu + r\sigma)$ is bounded by

$$\binom{k}{\lceil k/2 \rceil} \left(\frac{1}{r^2}\right)^{\lceil k/2 \rceil} < 2^k \left(\frac{1}{r^2}\right)^{\lceil k/2 \rceil} \leq \left(\frac{2}{r}\right)^k$$

$\square$

**Lemma A.2** (2.3). $P_2(\{\pm 1\}^d, d) \geq e^{d/8}$

*Proof of 2.3.* If $\mathbf{y}, \mathbf{x} \in \{\pm 1\}^d$ are two independent uniform vectors then by Hoeffding's bound we have

$$\Pr\left(\|\mathbf{x} - \mathbf{y}\|^2 \leq d\right) = \Pr\left(\sum_{i=1}^{d} 1[x_i \neq y_i] \leq d/4\right) \leq e^{-d/8}$$

Thus, there are at least $e^{d/8}$ vectors in $\{\pm 1\}^d$ such that the distance between each pair is more than $d$. $\square$

## A.2    MULTICOVER FOR VECTORS

**Lemma A.3** (2.5). *For the ball $B_M^d = \left\{\mathbf{x} \in \mathbb{R}^d \mid \|\mathbf{x}\|_2 \leq M\right\}$ and $\varepsilon \leq M$ we have*

$$2^{\min(d, \lfloor (M/2\varepsilon)^2 \rfloor)} \leq M_{\mathcal{R}_1^d}(B_M^d, \varepsilon) \leq \min\left((4d^2\lceil M \rceil + 6d)^{\lceil \frac{2M^2 + \frac{1}{4}}{\varepsilon^2} \rceil}, (3M/\varepsilon)^d\right)$$

*Proof of lemma 2.5.* We first show that $M_{\mathcal{R}_1^d}(B_M^d, \varepsilon) \leq (4d^2\lceil M \rceil + 6d)^{\lceil \frac{2M^2 + \frac{1}{4}}{\varepsilon^2} \rceil}$. By lemma 2.4, it is enough to show that there is a set $\mathcal{X} \subset \mathbb{R}^d$ of size $(4d^2\lceil M \rceil + 6d)^{\lceil \frac{2M^2 + \frac{1}{4}}{\varepsilon^2} \rceil}$ such that for every $\mathbf{x} \in B_M^d$ there is a random vector $X \in \mathcal{X}$ which is an $\varepsilon$-estimator of $\mathbf{x}$. Define

$$\tilde{\mathcal{X}} = \{k\mathbf{e}_i \mid i \in [d], k \in [-2dM - 1, 2dM + 1] \cap \mathbb{Z}\}$$

Let $\mathbf{x} \in B_M^d$, we next define a $\sqrt{2M^2 + \frac{1}{4}}$- estimator $X \in \tilde{\mathcal{X}}$ for $\mathbf{x}$: First sample a coordinate $i$ w.p. $p_i = \frac{\mathbf{x}_i^2}{2\|\mathbf{x}\|^2} + \frac{1}{2d}$, and let $\tilde{\mathbf{x}} = \left(\left\lfloor \frac{\mathbf{x}_i}{p_i} \right\rfloor + b\right) e_i$ where $b \sim Ber\left(\left\langle \frac{\mathbf{x}_i}{p_i} \right\rangle\right)$ and $\left\langle \frac{\mathbf{x}_i}{p_i} \right\rangle := \frac{\mathbf{x}_i}{p_i} - \left\lfloor \frac{\mathbf{x}_i}{p_i} \right\rfloor$.

Note that $\mathbb{E}X = \mathbf{x}$. Fix $\mathbf{u} \in \mathbb{S}^{d-1}$. We need to show that $\mathbb{E}\langle \mathbf{u}, X - \mathbf{x}\rangle^2 \leq 2M^2 + \frac{1}{4}$. Indeed,

$$
\begin{aligned}
\mathbb{E}\langle \mathbf{u}, X - \mathbf{x}\rangle^2 &\leq \mathbb{E}\langle \mathbf{u}, X\rangle^2 \\
&= \sum_i p_i \left( \left\langle \frac{\mathbf{x}_i}{p_i}\right\rangle \left( \left\lfloor \frac{\mathbf{x}_i}{p_i}\right\rfloor + 1\right)^2 + \left(1 - \left\langle \frac{\mathbf{x}_i}{p_i}\right\rangle\right)\left\lfloor \frac{\mathbf{x}_i}{p_i}\right\rfloor^2 \right)\mathbf{u}_i^2 \\
&= \sum_i p_i \left( 2\left\langle \frac{\mathbf{x}_i}{p_i}\right\rangle \left\lfloor \frac{\mathbf{x}_i}{p_i}\right\rfloor + \left\langle \frac{\mathbf{x}_i}{p_i}\right\rangle + \left\lfloor \frac{\mathbf{x}_i}{p_i}\right\rfloor^2 \right)\mathbf{u}_i^2 \\
&= \sum_i p_i \left( \left(\left\langle \frac{\mathbf{x}_i}{p_i}\right\rangle + \left\lfloor \frac{\mathbf{x}_i}{p_i}\right\rfloor\right)^2 + \left\langle \frac{\mathbf{x}_i}{p_i}\right\rangle\left(1 - \left\langle \frac{\mathbf{x}_i}{p_i}\right\rangle\right)\right)\mathbf{u}_i^2 \\
&= \sum_i p_i \left( \left(\frac{\mathbf{x}_i}{p_i}\right)^2 + \left\langle \frac{\mathbf{x}_i}{p_i}\right\rangle\left(1 - \left\langle \frac{\mathbf{x}_i}{p_i}\right\rangle\right)\right)\mathbf{u}_i^2 \\
&\leq \sum_i p_i \left( \left(\frac{\mathbf{x}_i}{p_i}\right)^2 + \frac{1}{4}\right)\mathbf{u}_i^2 \qquad (2) \\
&\leq \frac{1}{4}\|\mathbf{u}\|_\infty + \sum_i \frac{\mathbf{x}_i^2}{p_i}\mathbf{u}_i^2 \\
&\leq \frac{1}{4} + \sum_i 2\|\mathbf{x}\|^2 \mathbf{u}_i^2 \qquad (3) \\
&= 2\|\mathbf{x}\|^2 + \frac{1}{4} \\
&\leq 2M^2 + \frac{1}{4}
\end{aligned}
$$

Where equation 2 is true since $x(1-x) \leq 1/4$ for any $0 \leq x \leq 1$ and equation 3 is true by plugging in the definition of $p_i = \frac{\mathbf{x}_i^2}{2\|\mathbf{x}\|^2} + \frac{1}{2d}$, and by the fact that $\mathbf{u}$ is a unit vector.

We next construct an $\varepsilon$-estimator by averaging independent copies of $X$. Let $\tilde{X} = \frac{1}{k}\sum_{i=1}^k X_i$ where every $X_i$ is sampled i.i.d. like $X$. We claim that $\tilde{X}$ is $\sqrt{\frac{2M^2 + \frac{1}{4}}{k}}$-estimator. Let $\mathbf{u} \in \mathbb{S}^{d-1}$. We have that $\left\langle \mathbf{u}, \tilde{X} - \mathbf{x}\right\rangle = \frac{1}{k}\sum_{i=1}^k \langle \mathbf{u}, X_i - \mathbf{x}\rangle$ is an average of $k$ i.i.d. r.v. with mean 0 and variance bounded by $2M^2 + \frac{1}{4}$. Thus, $\mathbb{E}\left\langle \mathbf{u}, \tilde{X} - \mathbf{x}\right\rangle^2 \leq \frac{2M^2 + \frac{1}{4}}{k}$. Plugging $k = \lceil \frac{2M^2 + \frac{1}{4}}{\varepsilon^2}\rceil$, we get that $\mathbb{E}\left\langle \mathbf{u}, \tilde{X} - \mathbf{x}\right\rangle^2 \leq \varepsilon^2$. Note that $\tilde{X}$ gets values in $\mathcal{X} = \left\{ \frac{1}{k}\sum_{i=1}^k \mathbf{x}_i : \mathbf{x}_i \in \tilde{\mathcal{X}}\right\}$. By lemma 2.4 we have that $\mathcal{X}$ is a multicover. Finally, $|\mathcal{X}| \leq (4d^2\lceil M\rceil + 6d)^{\lceil \frac{2M^2 + \frac{1}{4}}{\varepsilon^2}\rceil}$, implying that $M_{\mathcal{R}_1^d}(B_M^d, \varepsilon) \leq (4d^2\lceil M\rceil + 6d)^{\lceil \frac{2M^2 + \frac{1}{4}}{\varepsilon^2}\rceil}$.

We next show that $M_{\mathcal{R}_1^d}(B_M^d, \varepsilon) \leq (3M/\varepsilon)^d$. Given Lemma 2.2, it is enough to show that $M_{\mathcal{R}_1^d}(B_M^d, \varepsilon) \leq N_2(B_M^d, \varepsilon)$. Indeed, fix $R \in \mathcal{R}_1^d$. We have for any $\mathbf{x}, \check{\mathbf{x}} \in \mathbb{R}^d$

$$\|\mathbf{x} - \check{\mathbf{x}}\|_R^2 = (\mathbf{x} - \check{\mathbf{x}})^\top R(\mathbf{x} - \check{\mathbf{x}}) \leq \|R\| \cdot \|\mathbf{x} - \check{\mathbf{x}}\|_2^2 \leq \mathrm{Tr}(R)\|\mathbf{x} - \check{\mathbf{x}}\|_2^2 \leq \|\mathbf{x} - \check{\mathbf{x}}\|_2^2$$

Thus, any $\varepsilon$-cover w.r.t. the Euclidean norm is an $\varepsilon$-cover w.r.t. $d_R$. Since this is true for any $R \in \mathcal{R}_1^d$, we have that any $\varepsilon$-cover w.r.t. the Euclidean norm is an $\varepsilon$-multicover. Thus, $M_{\mathcal{R}_1^d}(B_M^d, \varepsilon) \leq N_2(B_M^d, \varepsilon)$

we can use standard upper bounds for covering numbers of sets using volume (Vershynin, 2018) to upper bound the cover of $B_M^d$ with $O\left((3M/\varepsilon)^d\right)$ balls of radius $\varepsilon$ in $\ell_2$. This is an upper bound for the multicover of $B_M^d$ as well, considering that $\|\mathbf{x} - \check{\mathbf{x}}\|_R^2 = (\mathbf{x} - \check{\mathbf{x}})^\top R(\mathbf{x} - \check{\mathbf{x}}) \leq \mathrm{Tr}(R)(\mathbf{x} - \check{\mathbf{x}})^\top(\mathbf{x} - \check{\mathbf{x}}) = Tr(R)\|\mathbf{x} - \check{\mathbf{x}}\|_2^2$ where the inequality is by cauchy-schwarz and the fact that $\|x\|_2 \leq \|x\|_1$. Therefore we have shown the second upper bound.

For the lower bound, let $d' = \min\left(\lfloor (M/2\varepsilon)^2 \rfloor, d\right)$, and let $R = \frac{1}{d'}\tilde{I}_{d'}$ where $\tilde{I}_k$ is a diagonal matrix s.t. $\tilde{I}_{ii} = 0$ for $i > k$ and $\tilde{I}_{ii} = 1$ for $i \leq k$. We have

$$M_{\mathcal{R}_1^d}(B_M^d, \varepsilon) \geq N_R(B_M^d, \varepsilon) = N_2(B_M^{d'}, \sqrt{d'}\varepsilon) \geq N_2(B_M^{d'}, M/2) \overset{\text{Lemma 2.2}}{\geq} 2^{d'}$$

$\square$

**Lemma A.4** (2.6). *Let $\mathcal{S} \subset \mathcal{R}^d$ be a nice set, then:*

1. *For $\mathcal{X} \subset \mathbb{R}^{d_1}$ and a $d_2 \times d_1$ matrix $A$ we have $M_{\mathcal{S}}(A\mathcal{X}, \|A\|\varepsilon) \leq M(\mathcal{X}, \varepsilon)$*

2. *For $\mathcal{X}_1, \ldots, \mathcal{X}_n \subset \mathbb{R}^d$ and $\varepsilon_1, \ldots, \varepsilon_n > 0$ we have $M_{\mathcal{S}}(\sum_{i=1}^n \mathcal{X}_i, \sum_{i=1}^n \varepsilon_i) \leq \prod_{i=1}^n M_{\mathcal{S}}(\mathcal{X}_i, \varepsilon_i)$*

3. *For $\mathcal{X} \subset \mathbb{R}^d, \varepsilon > 0$, orthonormal matrix $U$ and $\mathbf{b} \in \mathbb{R}^d$ we have $M_{\mathcal{S}}(U\mathcal{X} + \mathbf{b}, \varepsilon) = M_{\mathcal{S}}(\mathcal{X}, \varepsilon)$*

4. *For $\mathcal{X}_1, \ldots, \mathcal{X}_n \subset \mathbb{R}^d$ and $\varepsilon > 0$ we have $M_{\mathcal{S}}(\cup_{i=1}^n \mathcal{X}_i, \varepsilon) \leq \sum_{i=1}^n M_{\mathcal{S}}(\mathcal{X}_i, \varepsilon)$*

5. *For $\mathcal{S} = \mathcal{R}_1^d$ and $\mathcal{X}_i \subset [-M_i, M_i]^d$ and $\varepsilon_1, \ldots, \varepsilon_n > 0$ we have*

$$M_{\mathcal{R}_1^d}\left(\prod_{i=1}^n \mathcal{X}_i, \prod_{i=1}^n (M_i + \varepsilon_i) - \prod_{i=1}^n M_i\right) \leq \prod_{i=1}^n M_{\mathcal{R}_1^d}(\mathcal{X}_i, \varepsilon_i)$$

6. *If for any maximal $R \in \mathcal{S}$ (w.r.t. PSD order), $Tr(R) \geq 1$. Fix $\mathcal{X} \subset B_M^{d_1}$, and $\mathcal{L} \subset \mathbb{R}^{d_2, d_1}$ matrices with spectral norm $\leq r$. Denote $\|A\|_{\mathcal{S}} := \min\{t > 0 : \frac{1}{t}A \in \mathcal{S}\}$, then*

$$M_{\mathcal{S}}\left(\mathcal{L}\mathcal{X}, \varepsilon_2\sqrt{2r^2 + 2\varepsilon_1^2\|I_{d_1}\|_{\mathcal{S}}} + \varepsilon_1 M\right) \leq M_{\mathcal{R}_1^d}(\mathcal{L}, \varepsilon_1) \cdot M_{\mathcal{S}}(\mathcal{X}, \varepsilon_2)$$

*Proof.* We next prove each item separately.

*Proof.* (of item 1.) Let $\check{\mathcal{X}}$ be an $\varepsilon$-multicover of $\mathcal{X}$ w.r.t. $\mathcal{S}$. It is enough to show that $A\check{\mathcal{X}}$ is an $(\|A\|\varepsilon)$-multicover of $A\mathcal{X}$. Fix $\mathcal{X} \in \mathcal{X}$ and a PSD matrix $R \in \mathcal{S}$. We need to show that there is $\check{\mathbf{x}} \in \check{\mathcal{X}}$ such that
$$\|A\mathbf{x} - A\check{\mathbf{x}}\|_R^2 \leq Tr(R)\|A\|^2\varepsilon^2$$
Now, for any $\check{\mathbf{x}}$ we have

$$\|A\mathbf{x} - A\check{\mathbf{x}}\|_R^2 = \|A\|^2(\mathbf{x} - \check{\mathbf{x}})^\top \frac{A^\top}{\|A\|} R \frac{A}{\|A\|}(\mathbf{x} - \check{\mathbf{x}}) = \|A\|^2\|\mathbf{x} - \check{\mathbf{x}}\|^2_{\frac{A^\top}{\|A\|} R \frac{A}{\|A\|}}$$

Finally, since $\check{\mathcal{X}}$ is an $\varepsilon$-multicover w.r.t. $\mathcal{S}$, there is $\check{\mathbf{x}} \in \check{\mathcal{X}}$ such that $\|\mathbf{x} - \check{\mathbf{x}}\|^2_{\frac{A^\top}{\|A\|} R \frac{A}{\|A\|}} \leq Tr(\frac{1}{\|A\|}A^\top R \frac{1}{\|A\|}A)\varepsilon^2 \leq Tr(R)\varepsilon^2$. Therefore overall

$$\|A\mathbf{x} - A\check{\mathbf{x}}\|_R^2 = \|A\|^2\|\mathbf{x} - \check{\mathbf{x}}\|^2_{\frac{A^\top}{\|A\|} R \frac{A}{\|A\|}} \leq Tr(R)\|A\|^2\varepsilon^2$$

$\square$

*Proof.* (of item 2.) Let $\check{\mathcal{X}}_i$ be an $\varepsilon_i$-multicover of $\mathcal{X}_i$ w.r.t. $\mathcal{S}$. It is not hard to verify that $\sum_{i=1}^n \check{\mathcal{X}}_i$ is an $(\sum_{i=1}^n \varepsilon_i)$-multicover of $\sum_{i=1}^n \mathcal{X}_i$, which establishes the proof. $\square$

*Proof.* (of item 3.) We have

$$M_{\mathcal{S}}(U\mathcal{X} + \mathbf{b}, \varepsilon) \overset{Item\ 2}{\leq} M_{\mathcal{S}}(U\mathcal{X}, \varepsilon) \cdot M_{\mathcal{S}}(\{\mathbf{b}\}, 0)$$
$$\overset{M_{\mathcal{S}}(\{\mathbf{b}\},0)=1,\ \|U\|=1}{=} M_{\mathcal{S}}(U\mathcal{X}, \|U\|\varepsilon)$$
$$\overset{Item\ 1}{\leq} M_{\mathcal{S}}(\mathcal{X}, \varepsilon)$$

Similarly $M_{\mathcal{S}}(\mathcal{X}, \varepsilon) = M_{\mathcal{S}}(U^{-1}(U\mathcal{X} + \mathbf{b}) - U^{-1}\mathbf{b}, \varepsilon) \leq M_{\mathcal{S}}(U\mathcal{X} + \mathbf{b}, \varepsilon)$ implying that $M_{\mathcal{S}}(\mathcal{X}, \varepsilon) = M_{\mathcal{S}}(U\mathcal{X} + \mathbf{b}, \varepsilon)$ $\square$

*Proof.* (of item 4.) Let $\check{\mathcal{X}}_i$ be an $\varepsilon$-multicover of $\mathcal{X}_i$ w.r.t. $\mathcal{S}$. It is not hard to verify that $\cup_{i=1}^n \check{\mathcal{X}}_i$ is an $\varepsilon$-multicover of $\cup_{i=1}^n \mathcal{X}_i$ w.r.t. $\mathcal{S}$, which establishes the proof. $\qquad\square$

*Proof.* (of item 5.) We first prove the item for $n = 2$. We will then show that the general case follows by induction. In the proof of this item we will denote by $A \circ B$ the elementwise product of two $d \times d$ matrices, and by $\mathrm{diag}(A)$ the diagonal matrix obtained by zeroing the non-diagonal entries of $A$.

Let $\check{\mathcal{X}}_i$ be an $\varepsilon_i$-multicover of $\mathcal{X}_i$ w.r.t. $\mathcal{R}_1^d$. Fix $\mathbf{x}_i \in \mathcal{X}_i$ and a PSD matrix $R \geq 0$ with $\mathrm{Tr}(R) \leq 1$. It is enough to show that there is $\check{\mathbf{x}}_i \in \check{\mathcal{X}}_i$ with $\|\mathbf{x}_1\mathbf{x}_2 - \check{\mathbf{x}}_1\check{\mathbf{x}}_2\|_R \leq M_1\varepsilon_2 + M_2\varepsilon_1 + \varepsilon_1\varepsilon_2$. We have

$$
\begin{aligned}
\|\mathbf{x}_1\mathbf{x}_2 - \check{\mathbf{x}}_1\check{\mathbf{x}}_2\|_R &\leq \|\mathbf{x}_1\mathbf{x}_2 - \mathbf{x}_1\check{\mathbf{x}}_2\|_R + \|\mathbf{x}_1\check{\mathbf{x}}_2 - \check{\mathbf{x}}_1\check{\mathbf{x}}_2\|_R \\
&= \|\mathbf{x}_2 - \check{\mathbf{x}}_2\|_{R\circ\mathbf{x}_1\mathbf{x}_1^\top} + \|\mathbf{x}_1 - \check{\mathbf{x}}_1\|_{R\circ\check{\mathbf{x}}_2\check{\mathbf{x}}_2^\top}
\end{aligned}
$$

Now, $\mathrm{Tr}(R \circ \check{\mathbf{x}}_2\check{\mathbf{x}}_2^\top) = \|\check{\mathbf{x}}_2\|_{\mathrm{diag}(R)}^2$. Thus, we can choose $\check{\mathbf{x}}_1$ such that $\|\mathbf{x}_1 - \check{\mathbf{x}}_1\|_{R\circ\check{\mathbf{x}}_2\check{\mathbf{x}}_2^\top} \leq \|\check{\mathbf{x}}_2\|_{\mathrm{diag}(R)}\varepsilon_1$. We get for any $0 < p < 1$

$$
\begin{aligned}
\|\mathbf{x}_1\mathbf{x}_2 - \check{\mathbf{x}}_1\check{\mathbf{x}}_2\|_R &\leq \|\mathbf{x}_2 - \check{\mathbf{x}}_2\|_{R\circ\mathbf{x}_1\mathbf{x}_1^\top} + \|\check{\mathbf{x}}_2\|_{\mathrm{diag}(R)}\varepsilon_1 \\
&\leq \|\mathbf{x}_2 - \check{\mathbf{x}}_2\|_{R\circ\mathbf{x}_1\mathbf{x}_1^\top} + \|\check{\mathbf{x}}_2 - \mathbf{x}_2\|_{\mathrm{diag}(R)}\varepsilon_1 + \|\mathbf{x}_2\|_{\mathrm{diag}(R)}\varepsilon_1 \\
&\leq \|\mathbf{x}_2 - \check{\mathbf{x}}_2\|_{R\circ\mathbf{x}_1\mathbf{x}_1^\top} + \|\check{\mathbf{x}}_2 - \mathbf{x}_2\|_{\mathrm{diag}(R)}\varepsilon_1 + M_2\varepsilon_1 \\
&= \|\mathbf{x}_2 - \check{\mathbf{x}}_2\|_{R\circ\mathbf{x}_1\mathbf{x}_1^\top} + \|\check{\mathbf{x}}_2 - \mathbf{x}_2\|_{\mathrm{diag}(\varepsilon_1^2 R)} + M_2\varepsilon_1 \\
&\overset{(*)}{\leq} \sqrt{\frac{\|\mathbf{x}_2 - \check{\mathbf{x}}_2\|_{R\circ\mathbf{x}_1\mathbf{x}_1^\top}^2}{p} + \frac{\|\check{\mathbf{x}}_2 - \mathbf{x}_2\|_{\mathrm{diag}(\varepsilon_1^2 R)}^2}{1 - p}} + M_2\varepsilon_1 \\
&= \|\mathbf{x}_2 - \check{\mathbf{x}}_2\|_{\frac{1}{p}R\circ\mathbf{x}_1\mathbf{x}_1^\top + \frac{1}{1-p}\mathrm{diag}(\varepsilon_1^2 R)} + M_2\varepsilon_1
\end{aligned}
$$

Where (*) follows from the fact that $a + b \leq \sqrt{a^2/p + b^2/(1-p)}$ Now, we can choose $\check{\mathbf{x}}_2 \in \check{\mathcal{X}}_2$ with

$$
\begin{aligned}
\|\mathbf{x}_2 - \check{\mathbf{x}}_2\|_{\frac{1}{p}R\circ\mathbf{x}_1\mathbf{x}_1^\top + \frac{1}{1-p}\mathrm{diag}(\varepsilon_1^2 R)} &\leq \varepsilon_2\sqrt{\mathrm{Tr}\left(\frac{1}{p}R \circ \mathbf{x}_1\mathbf{x}_1^\top + \frac{1}{1-p}\mathrm{diag}(\varepsilon_1^2 R)\right)} \\
&\leq \varepsilon_2\sqrt{M_1^2/p + \varepsilon_1^2/(1-p)}
\end{aligned}
$$

for $p = M_1/(M_1 + \varepsilon_1)$ we get that

$$
\|\mathbf{x}_1\mathbf{x}_2 - \check{\mathbf{x}}_1\check{\mathbf{x}}_2\|_R \leq \varepsilon_2(M_1 + \varepsilon_1) + M_2\varepsilon_1 = M_1\varepsilon_2 + M_2\varepsilon_1 + \varepsilon_1\varepsilon_2
$$

We next consider $n > 2$ and conclude the proof by induction. Denote $\mathcal{X}_2' = \prod_{i=2}^n \mathcal{X}_i$, $M_2' = \prod_{i=2}^n M_i$ and $\varepsilon_2' = \prod_{i=2}^n (M_i + \varepsilon_i) - \prod_{i=2}^n M_i$. By the induction hypothesis we have

$$
M(\mathcal{X}_2', \varepsilon_2') \leq \prod_{i=2}^n M(\mathcal{X}_i, \varepsilon_i)
$$

By the case $n = 2$ we have

$$
\begin{aligned}
M(\mathcal{X}_1\mathcal{X}_2', M_1\varepsilon_2' + M_2'\varepsilon_1 + \varepsilon_1\varepsilon_2') &\leq M(\mathcal{X}_1, \varepsilon_1) M(\mathcal{X}_2', \varepsilon_2') \\
&\leq M(\mathcal{X}_1, \varepsilon_1) \prod_{i=2}^n M(\mathcal{X}_i, \varepsilon_i) = \prod_{i=2}^n M(\mathcal{X}_i, \varepsilon_i)
\end{aligned}
$$

this concludes the proof as $\mathcal{X}_1\mathcal{X}_2' = \prod_{i=1}^n \mathcal{X}_i$ and

$$
\begin{aligned}
M_1\varepsilon_2' + M_2'\varepsilon_1 + \varepsilon_1\varepsilon_2' &= (M_1 + \varepsilon_1)\left(\prod_{i=2}^n (M_i + \varepsilon_i) - \prod_{i=2}^n M_i\right) + \varepsilon_1 \prod_{i=2}^n M_i \\
&= \prod_{i=1}^n (M_i + \varepsilon_i) - \prod_{i=1}^n M_i
\end{aligned}
$$

$\qquad\square$

*Proof.* (of item 6) For a nice set $\mathcal{S} \subset \mathcal{R}^d$, and PSD matrix $R \in \mathbb{R}^{d \times d}$, define $\|R\|_{\mathcal{S}} = \min\{t > 0 : \frac{1}{t}R \in \mathcal{S}\}$. Note that this is almost a norm - the triangle inequality, positive definiteness, and homogeneity for positive scalars apply - but do not apply for negative scalars. Let $\check{\mathcal{L}}$ be an $\varepsilon_1$-multicover of $\mathcal{L}$ w.r.t. $\mathcal{R}_1^d$ and let $\check{\mathcal{X}}$ be an $\varepsilon_2$-multicover of $\mathcal{X}$ w.r.t. $\mathcal{S}$. We will show that $\check{\mathcal{L}}\check{\mathcal{X}}$ is an $\varepsilon_2\sqrt{2r^2 + 2\varepsilon_1\|I_{d_1}\|_{\mathcal{S}}} + \varepsilon_1 M$-multicover of $\mathcal{L}\mathcal{X}$ w.r.t. $\mathcal{S}$. Fix $R \in \mathcal{S}$. W.l.o.g we may assume that it is maximal w.r.t. PSD order. Let $W \in \mathcal{L}$ and $\mathbf{x} \in \mathcal{X}$. We need to show that there are $\check{W} \in \check{\mathcal{L}}$ and $\check{\mathbf{x}} \in \check{\mathcal{X}}$ with $\|W\mathbf{x} - \check{W}\check{\mathbf{x}}\|_R \le \varepsilon_2\sqrt{2\mathrm{Tr}(R)r^2 + 2\varepsilon_1\mathrm{Tr}(R)\|I_{d_1}\|_{\mathcal{S}}} + \varepsilon_1\sqrt{\mathrm{Tr}(R)}M$. We have

$$
\begin{aligned}
\|W\mathbf{x} - \check{W}\check{\mathbf{x}}\|_R &\le \|W\mathbf{x} - W\check{\mathbf{x}}\|_R + \|W\check{\mathbf{x}} - \check{W}\check{\mathbf{x}}\|_R \\
&= \|\mathbf{x} - \check{\mathbf{x}}\|_{W^\top RW} + \|(W - \check{W})\check{\mathbf{x}}\|_R \\
&= \|\mathbf{x} - \check{\mathbf{x}}\|_{W^\top RW} + \sqrt{\check{\mathbf{x}}^\top(W - \check{W})^\top R(W - \check{W})\check{\mathbf{x}}}
\end{aligned}
$$

Now, $(W_1, W_2) \mapsto \check{\mathbf{x}}^\top W_1^\top RW_2\check{\mathbf{x}}$ is a symmetric and positive bi-linear form on the space of $d_2 \times d_1$ matrices of trace

$$
\sum_{i=1}^{d_2}\sum_{j=1}^{d_1} \check{\mathbf{x}}^\top E_{ij}^\top RE_{ij}\check{\mathbf{x}} = \sum_{i=1}^{d_2}\sum_{j=1}^{d_1}(\check{x}_j\mathbf{e}_i)^\top R(\check{x}_j\mathbf{e}_i) = \sum_{i=1}^{d_2}\sum_{j=1}^{d_1}\check{x}_j^2 R_{ii} = \mathrm{Tr}(R)\|\check{\mathbf{x}}\|^2
$$

Thus, there is $\check{W} \in \check{\mathcal{L}}$ such that $\check{\mathbf{x}}^\top(W - \check{W})^\top R(W - \check{W})\check{\mathbf{x}} \le \mathrm{Tr}(R)\|\check{\mathbf{x}}\|^2\varepsilon_1^2$. For this $\check{W}$ we have

$$
\begin{aligned}
\|W\mathbf{x} - \check{W}\check{\mathbf{x}}\|_R &\le \|\mathbf{x} - \check{\mathbf{x}}\|_{W^\top RW} + \varepsilon_1\sqrt{\mathrm{Tr}(R)}\|\check{\mathbf{x}}\| \\
&\le \|\mathbf{x} - \check{\mathbf{x}}\|_{W^\top RW} + \varepsilon_1\sqrt{\mathrm{Tr}(R)}(\|\check{\mathbf{x}} - \mathbf{x}\| + \|\mathbf{x}\|) \\
&\le \|\mathbf{x} - \check{\mathbf{x}}\|_{W^\top RW} + \|\check{\mathbf{x}} - \mathbf{x}\|_{\varepsilon_1^2\mathrm{Tr}(R)I_{d_1}} + \varepsilon_1\sqrt{\mathrm{Tr}(R)}M \\
&\le \sqrt{2}\sqrt{\|\mathbf{x} - \check{\mathbf{x}}\|_{W^\top RW}^2 + \|\check{\mathbf{x}} - \mathbf{x}\|_{\varepsilon_1^2\mathrm{Tr}(R)I_{d_1}}^2} + \varepsilon_1\sqrt{\mathrm{Tr}(R)}M \\
&= \sqrt{2}\|\mathbf{x} - \check{\mathbf{x}}\|_{W^\top RW + \varepsilon_1^2\mathrm{Tr}(R)I_{d_1}} + \varepsilon_1\sqrt{\mathrm{Tr}(R)}M
\end{aligned}
$$

Thus, it is possible to choose $\check{\mathbf{x}} \in \check{\mathcal{X}}$ s.t. $\|\mathbf{x} - \check{\mathbf{x}}\|_{W^\top RW + \varepsilon_1^2 I_{d_1}} \le \varepsilon_2\sqrt{\|W^\top RW + \varepsilon_1^2\mathrm{Tr}(R)I_{d_1}\|_{\mathcal{S}}}$. Finally, $\|W^\top RW + \varepsilon_1^2\mathrm{Tr}(R)I_{d_1}\|_{\mathcal{S}} \le r^2 + \varepsilon_1^2\mathrm{Tr}(R)\|I_{d_1}\|_{\mathcal{S}} \le Tr(R)r^2 + \varepsilon_1^2\mathrm{Tr}(R)\|I_{d_1}\|_{\mathcal{S}}$  □

□

### A.3 MULTICOVER FOR SEQUENCES OF VECTORS

*Proof.* (of Lemma 3.1) Write $\check{\mathcal{X}} = \{\mathbf{x}^1, \ldots, \mathbf{x}^T\}$. Suppose that $\check{\mathcal{X}}$ is a $\varepsilon$-multicover w.r.t. $\mathcal{R}^{d,m}$ and let $\mathbf{x} \in \mathcal{X}$. It is enough to show that there is a r.v. $X$ whose range is $\{\mathbf{x}^1, \ldots, \mathbf{x}^T\}$ such that for any $R \in \mathcal{R}_1^{d,m}$, $\mathbb{E}\|X - \mathbf{x}\|_R^2 \le \varepsilon^2$. Such a r.v. exists if and only if

$$
\min_{\lambda \in \Delta^{T-1}} \max_{R \in \mathcal{R}_1^{d,m}} \sum_{i=1}^T \lambda_i\|\mathbf{x}^i - \mathbf{x}\|_R^2 \le \varepsilon^2
$$

since the objective $\sum_{i=1}^T \lambda_i\|\mathbf{x}^i - \mathbf{x}\|_R^2 = \sum_{i=1}^T\sum_{m=1}^m \lambda_i(\mathbf{x}_j^i - \mathbf{x}_j)^\top R(\mathbf{x}_j^i - \mathbf{x}_j)$ is bi-linear in $\lambda$ and $R$, and since $\Delta^{T-1}$ and $\mathcal{R}_1^{d,m}$ are both convex and compact, we can apply the minmax theorem to conclude that a r.v. $X$ as described above exists if and only if

$$
\max_{R \in \mathcal{R}_1^{d,m}} \min_{\lambda \in \Delta^{T-1}} \sum_{i=1}^T \lambda_i\|\mathbf{x}^i - \mathbf{x}\|_R^2 \le \varepsilon^2
$$

which is equivalent to $\max_{R \in \mathcal{R}_1^d} \min_{i \in [T]} \|\mathbf{x}^i - \mathbf{x}\|_R \le \varepsilon$. Which is indeed the case as $\check{\mathcal{X}}$ is an $\varepsilon$-multicover on $\mathcal{X}$ w.r.t. $\mathcal{R}^{d,m}$.

Suppose now that for any $\mathbf{x} \in \mathcal{X}$ there is a r.v. $X$ whose range is $\check{\mathcal{X}}$ such that for any $R \in \mathcal{R}_1^{d,m}$, $\mathbb{E}\|X - \mathbf{x}\|_R^2 \le \varepsilon^2$. This implies that for any $\mathbf{x} \in \mathcal{X}$ and any $R \in \mathcal{R}_1^d$ there is $\check{\mathbf{x}} \in \check{\mathcal{X}}$ such that $\|\check{\mathbf{x}} - \mathbf{x}\|_R \le \varepsilon$. This implies that $\check{\mathcal{X}}$ is an $\varepsilon$-multicover of $\mathcal{X}$ w.r.t. $\mathcal{R}^{d,m}$.  □

**Lemma A.5** (3.2).      *1. For $\mathcal{X} \subset \mathbb{R}^{d_1,m}$ and a $d_2 \times d_1$ matrix $A$ we have $M_\mathcal{S}(A\mathcal{X}, \|A\|\varepsilon) \leq M_\mathcal{S}(\mathcal{X}, \varepsilon)$*

*2. For $\mathcal{X}_1, \ldots, \mathcal{X}_n \subset \mathbb{R}^{d,m}$ and $\varepsilon_1, \ldots, \varepsilon_n > 0$ we have $M_\mathcal{S}(\sum_{i=1}^n \mathcal{X}_i, \sum_{i=1}^n \varepsilon_i) \leq \prod_{i=1}^n M_\mathcal{S}(\mathcal{X}_i, \varepsilon_i)$*

*3. For $\mathcal{X} \subset \mathbb{R}^{d,m}, \varepsilon > 0$ and $\mathbf{b} \in \mathbb{R}^{d,m}$ we have $M_\mathcal{S}(U\mathcal{X} + \mathbf{b}, \varepsilon) = M_\mathcal{S}(\mathcal{X}, \varepsilon)$*

*4. For $\mathcal{X}_1, \ldots, \mathcal{X}_n \subset \mathbb{R}^{d,m}$ and $\varepsilon > 0$ we have $M_\mathcal{S}(\cup_{i=1}^n \mathcal{X}_i, \varepsilon) \leq \sum_{i=1}^n M_\mathcal{S}(\mathcal{X}_i, \varepsilon)$*

*5. For $\mathcal{S} = \mathcal{R}_1^{d,m} \, \mathcal{X}_i \subset [-M_i, M_i]^{d,m}$ and $\varepsilon_1, \ldots, \varepsilon_n > 0$ we have*

$$M_{\mathcal{R}_1^{d,m}}\left( \prod_{i=1}^n \mathcal{X}_i, \prod_{i=1}^n (M_i + \varepsilon_i) - \prod_{i=1}^n M_i \right) \leq \prod_{i=1}^n M_{\mathcal{R}_1^{d,m}}(\mathcal{X}_i, \varepsilon_i)$$

*6. For[3] $\mathcal{S} = \mathcal{R}_1^{d,m}$, fix $\mathcal{X} \subset B_M^{d_1,m}$ a set $\mathcal{L} \subset \mathbb{R}^{d_2,d_1}$ of matrices with spectral norm at most $r$. We have*

$$M_\mathcal{S}\left( \mathcal{L}\mathcal{X}, \varepsilon_2 \sqrt{2r^2 + 2\varepsilon_1^2 d_1} + \varepsilon_1 M \right) \leq M_{\mathcal{R}_1^{d,m}}(\mathcal{L}, \varepsilon_1) \cdot M_\mathcal{S}(\mathcal{X}, \varepsilon_2)$$

*Proof.* (of Lemma 3.2) Fix $R, S \in \mathcal{R}^{d,m}$, $\mathbf{x} \in \mathbb{R}^{d,m}$, $A \in \mathbb{R}^{d_1 \times d}$ and $B \in \mathbb{R}^{d \times d_2}$. In this proof we will denote $R \circ S = (R^1 \circ S^1, \ldots, R^m \circ S^m)$ where $R^j \circ S^j$ the elementwise product of $R^j$ and $S^j$. We will also denote $\text{diag}(R) = (\text{diag}(R^1), \ldots, \text{diag}(R^m))$, $AR = (AR^1, \ldots, AR^m)$, $RB = (R^1 B, \ldots, R^m B)$ and $\mathbf{x}\mathbf{x}^\top = (\mathbf{x}^1(\mathbf{x}^1)^\top, \ldots, \mathbf{x}^m(\mathbf{x}^m)^\top)$.

We next prove each item separately.

*Proof.* (of item 1.) Let $\check{\mathcal{X}}$ be an $\varepsilon$-multicover of $\mathcal{X}$ w.r.t. $\mathcal{S}$. It is enough to show that $A\check{\mathcal{X}}$ is an $(\|A\|\varepsilon)$-multicover of $A\mathcal{X}$ w.r.t. $\mathcal{S}$. Fix $\mathbf{x} \in \mathcal{X}$ and $R \in \mathcal{R}^{d,m}$. We need to show that there is $\check{\mathbf{x}} \in \check{\mathcal{X}}$ such that

$$\|A\mathbf{x} - A\check{\mathbf{x}}\|_R^2 \leq \text{Tr}(R)\|A\|^2 \varepsilon^2$$

Now, for any $\check{\mathbf{x}}$ we have

$$\|A\mathbf{x} - A\check{\mathbf{x}}\|_R^2 = \sum_{i=1}^n (\mathbf{x}_i - \check{\mathbf{x}}_i)^\top A^\top R_i A(\mathbf{x}_i - \check{\mathbf{x}}_i) = \|A\|^2 \|\mathbf{x} - \check{\mathbf{x}}\|_{\frac{A^\top}{\|A\|} R \frac{A}{\|A\|}}^2$$

Finally, since $\check{\mathcal{X}}$ is an $\varepsilon$-multicover, there is $\check{\mathbf{x}} \in \check{\mathcal{X}}$ such that $\|\mathbf{x} - \check{\mathbf{x}}\|_{\frac{A^\top}{\|A\|} R \frac{A}{\|A\|}}^2 \leq \text{Tr}\left( \frac{A^\top}{\|A\|} R \frac{A}{\|A\|} \right) \varepsilon^2 \leq \text{Tr}(R)\varepsilon^2$. Plugging in $\|A\mathbf{x} - A\check{\mathbf{x}}\|_R^2 = \|A\|^2 \|\mathbf{x} - \check{\mathbf{x}}\|_{\frac{A^\top}{\|A\|} R \frac{A}{\|A\|}}^2$ we establish the proof.

$\square$

*Proof.* (of item 2.) Let $\check{\mathcal{X}}_i$ be an $\varepsilon_i$-multicover of $\mathcal{X}_i$ w.r.t. $\mathcal{S}$. It is not hard to verify that $\sum_{i=1}^n \check{\mathcal{X}}_i$ is an $(\sum_{i=1}^n \varepsilon_i)$-multicover of $\sum_{i=1}^n \mathcal{X}_i$ w.r.t. $\mathcal{S}$, which establishes the proof. $\square$

*Proof.* (of item 3.) We have

$$M_\mathcal{S}(\mathcal{X} + \mathbf{b}, \varepsilon) \overset{Item\ 2}{\leq} M_\mathcal{S}(\mathcal{X}, \varepsilon) \cdot M_\mathcal{S}(\{\mathbf{b}\}, 0)$$
$$= M_\mathcal{S}(\mathcal{X}, \varepsilon)$$

Similarly $M_\mathcal{S}(\mathcal{X}, \varepsilon) = M_\mathcal{S}((\mathcal{X} + \mathbf{b}) - \mathbf{b}, \varepsilon) \leq M_\mathcal{S}(\mathcal{X} + \mathbf{b}, \varepsilon)$ implying that $M_\mathcal{S}(\mathcal{X}, \varepsilon) = M_\mathcal{S}(\mathcal{X} + \mathbf{b}, \varepsilon)$ $\square$

*Proof.* (of item 4.) Let $\check{\mathcal{X}}_i$ be an $\varepsilon$-multicover of $\mathcal{X}_i$ w.r.t. $\mathcal{S}$. It is not hard to verify that $\cup_{i=1}^n \check{\mathcal{X}}_i$ is an $\varepsilon$-multicover of $\cup_{i=1}^n \mathcal{X}_i$ w.r.t. $\mathcal{S}$, which establishes the proof. $\square$

---

[3]This claim can be generalized to a more general $\mathcal{S}$. We present the case $\mathcal{S} = \mathcal{R}_1^{d,m}$ for simplicity.

*Proof.* (of item 5.) We first prove the item for $n = 2$. We will then show that the general case follows by induction. Let $\check{\mathcal{X}}_i$ be an $\varepsilon_i$-multicover of $\mathcal{X}_i$. Fix $\mathbf{x}_i \in \mathcal{X}_i$ and a $R \in \mathcal{R}_1^{d,m}$. It is enough to show that there is $\check{\mathbf{x}}_i \in \check{\mathcal{X}}_i$ with $\|\mathbf{x}_1\mathbf{x}_2 - \check{\mathbf{x}}_1\check{\mathbf{x}}_2\|_R \le M_1\varepsilon_2 + M_2\varepsilon_1 + \varepsilon_1\varepsilon_2$. We have

$$
\begin{aligned}
\|\mathbf{x}_1\mathbf{x}_2 - \check{\mathbf{x}}_1\check{\mathbf{x}}_2\|_R &\le \|\mathbf{x}_1\mathbf{x}_2 - \mathbf{x}_1\check{\mathbf{x}}_2\|_R + \|\mathbf{x}_1\check{\mathbf{x}}_2 - \check{\mathbf{x}}_1\check{\mathbf{x}}_2\|_R \\
&= \|\mathbf{x}_2 - \check{\mathbf{x}}_2\|_{R\circ\mathbf{x}_1\mathbf{x}_1^\top} + \|\mathbf{x}_1 - \check{\mathbf{x}}_1\|_{R\circ\check{\mathbf{x}}_2\check{\mathbf{x}}_2^\top}
\end{aligned}
$$

Now, $\mathrm{Tr}(R \circ \check{\mathbf{x}}_2\check{\mathbf{x}}_2^\top) = \|\check{\mathbf{x}}_2\|_{\mathrm{diag}(R)}^2$. Thus, we can choose $\check{\mathbf{x}}_1$ such that $\|\mathbf{x}_1 - \check{\mathbf{x}}_1\|_{R\circ\check{\mathbf{x}}_2\check{\mathbf{x}}_2^\top} \le \|\check{\mathbf{x}}_2\|_{\mathrm{diag}(R)}\varepsilon_1$. We get for any $0 < p < 1$

$$
\begin{aligned}
\|\mathbf{x}_1\mathbf{x}_2 - \check{\mathbf{x}}_1\check{\mathbf{x}}_2\|_R &\le \|\mathbf{x}_2 - \check{\mathbf{x}}_2\|_{R\circ\mathbf{x}_1\mathbf{x}_1^\top} + \|\check{\mathbf{x}}_2\|_{\mathrm{diag}(R)}\varepsilon_1 \\
&\le \|\mathbf{x}_2 - \check{\mathbf{x}}_2\|_{R\circ\mathbf{x}_1\mathbf{x}_1^\top} + \|\check{\mathbf{x}}_2 - \mathbf{x}_2\|_{\mathrm{diag}(R)}\varepsilon_1 + \|\mathbf{x}_2\|_{\mathrm{diag}(R)}\varepsilon_1 \\
&\le \|\mathbf{x}_2 - \check{\mathbf{x}}_2\|_{R\circ\mathbf{x}_1\mathbf{x}_1^\top} + \|\check{\mathbf{x}}_2 - \mathbf{x}_2\|_{\mathrm{diag}(R)}\varepsilon_1 + M_2\varepsilon_1 \\
&= \|\mathbf{x}_2 - \check{\mathbf{x}}_2\|_{R\circ\mathbf{x}_1\mathbf{x}_1^\top} + \|\check{\mathbf{x}}_2 - \mathbf{x}_2\|_{\mathrm{diag}(\varepsilon_1^2 R)} + M_2\varepsilon_1 \\
&\overset{a+b\le\sqrt{a^2/p+b^2/(1-p)}}{\le} \sqrt{\frac{\|\mathbf{x}_2 - \check{\mathbf{x}}_2\|_{R\circ\mathbf{x}_1\mathbf{x}_1^\top}^2}{p} + \frac{\|\check{\mathbf{x}}_2 - \mathbf{x}_2\|_{\mathrm{diag}(\varepsilon_1^2 R)}^2}{1 - p}} + M_2\varepsilon_1 \\
&= \|\mathbf{x}_2 - \check{\mathbf{x}}_2\|_{\frac{1}{p}R\circ\mathbf{x}_1\mathbf{x}_1^\top + \frac{1}{1-p}\mathrm{diag}(\varepsilon_1^2 R)} + M_2\varepsilon_1
\end{aligned}
$$

Now, we can choose $\check{\mathbf{x}}_2 \in \check{\mathcal{X}}_2$ with

$$
\|\mathbf{x}_2 - \check{\mathbf{x}}_2\|_{\frac{1}{p}R\circ\mathbf{x}_1\mathbf{x}_1^\top + \frac{1}{1-p}\mathrm{diag}(\varepsilon_1^2 R)} \le \varepsilon_2\sqrt{\mathrm{Tr}\left(\frac{1}{p}R\circ\mathbf{x}_1\mathbf{x}_1^\top + \frac{1}{1-p}\mathrm{diag}(\varepsilon_1^2 R)\right)} \le \varepsilon_2\sqrt{M_1^2/p + \varepsilon_1^2/(1-p)}
$$

for $p = M_1/(M_1 + \varepsilon_1)$ we get that

$$
\|\mathbf{x}_1\mathbf{x}_2 - \check{\mathbf{x}}_1\check{\mathbf{x}}_2\|_R \le \varepsilon_2(M_1 + \varepsilon_1) + M_2\varepsilon_1 = M_1\varepsilon_2 + M_2\varepsilon_1 + \varepsilon_1\varepsilon_2
$$

We next consider $n > 2$ and conclude the proof by induction. Denote $\mathcal{X}_2' = \prod_{i=2}^n \mathcal{X}_i$, $M_2' = \prod_{i=2}^n M_i$ and $\varepsilon_2' = \prod_{i=2}^n (M_i + \varepsilon_i) - \prod_{i=2}^n M_i$. By the induction hypothesis we have

$$
M_{\mathcal{R}_1^{d,m}}(\mathcal{X}_2', \varepsilon_2') \le \prod_{i=2}^n M_{\mathcal{R}_1^{d,m}}(\mathcal{X}_i, \varepsilon_i)
$$

By the case $n = 2$ we have

$$
\begin{aligned}
M_{\mathcal{R}_1^{d,m}}(\mathcal{X}_1\mathcal{X}_2', M_1\varepsilon_2' + M_2'\varepsilon_1 + \varepsilon_1\varepsilon_2') &\le M_{\mathcal{R}_1^{d,m}}(\mathcal{X}_1, \varepsilon_1) M_{\mathcal{R}_1^{d,m}}(\mathcal{X}_2', \varepsilon_2') \\
&\le M_{\mathcal{R}_1^{d,m}}(\mathcal{X}_1, \varepsilon_1) \prod_{i=2}^n M_{\mathcal{R}_1^{d,m}}(\mathcal{X}_i, \varepsilon_i) = \prod_{i=2}^n M_{\mathcal{R}_1^{d,m}}(\mathcal{X}_i, \varepsilon_i)
\end{aligned}
$$

this concludes the proof as $\mathcal{X}_1\mathcal{X}_2' = \prod_{i=1}^n \mathcal{X}_i$ and

$$
\begin{aligned}
M_1\varepsilon_2' + M_2'\varepsilon_1 + \varepsilon_1\varepsilon_2' &= (M_1 + \varepsilon_1)\left(\prod_{i=2}^n (M_i + \varepsilon_i) - \prod_{i=2}^n M_i\right) + \varepsilon_1 \prod_{i=2}^n M_i \\
&= \prod_{i=1}^n (M_i + \varepsilon_i) - \prod_{i=1}^n M_i
\end{aligned}
$$

$\square$

*Proof.* (of item 6) Let $\mathcal{S} = \mathcal{R}_1^{d,m}$, let $\check{\mathcal{L}}$ be an $\varepsilon_1$-multicover of $\mathcal{L}$ w.r.t. $\mathcal{R}_1^{d,m}$ and let $\check{\mathcal{X}}$ be an $\varepsilon_2$-multicover of $\mathcal{X}$ w.r.t. $\mathcal{S}$. We will show that $\check{\mathcal{L}}\check{\mathcal{X}}$ is an $(\varepsilon_2\sqrt{2r^2 + 2\varepsilon_1^2 d_1} + \varepsilon_1 M)$-multicover of

$\mathcal{L}\mathcal{X}$ w.r.t. $\mathcal{S}$. Fix $R \in \mathcal{R}_1^{d,m}$. Let $W \in \mathcal{L}$ and $\mathbf{x} \in \mathcal{X}$. We need to show that there are $\check{W} \in \check{\mathcal{L}}$ and $\check{\mathbf{x}} \in \check{\mathcal{X}}$ with $\|W\mathbf{x} - \check{W}\check{\mathbf{x}}\|_R \leq \varepsilon_2\sqrt{2r^2 + 2\varepsilon_1^2 d_1} + \varepsilon_1 M$. We have

$$
\begin{aligned}
\|W\mathbf{x} - \check{W}\check{\mathbf{x}}\|_R &\leq \|W\mathbf{x} - W\check{\mathbf{x}}\|_R + \|W\check{\mathbf{x}} - \check{W}\check{\mathbf{x}}\|_R \\
&= \|\mathbf{x} - \check{\mathbf{x}}\|_{W^\top RW} + \|(W - \check{W})\check{\mathbf{x}}\|_R \\
&= \|\mathbf{x} - \check{\mathbf{x}}\|_{W^\top RW} + \sqrt{\sum_{i=1}^m (\check{\mathbf{x}}^i)^\top (W - \check{W})^\top R^i (W - \check{W})\check{\mathbf{x}}^i}
\end{aligned}
$$

Now, $(W_1, W_2) \mapsto \sum_{i=1}^m (\check{\mathbf{x}}^i)^\top W_1^\top R^i W_2 \check{\mathbf{x}}^i$ is a symmetric and positive bi-linear form on the space of $d_2 \times d_1$ matrices of trace

$$
\sum_{k=1}^m \sum_{i=1}^{d_2} \sum_{j=1}^{d_1} (\check{\mathbf{x}}^k)^\top E_{ij}^\top R^k E_{ij}\check{\mathbf{x}}^k = \sum_{k=1}^m \sum_{i=1}^{d_2} \sum_{j=1}^{d_1} (\check{x}_j^k \mathbf{e}_i)^\top R^k (\check{x}_j^k \mathbf{e}_i) = \sum_{k=1}^m \sum_{i=1}^{d_2} \sum_{j=1}^{d_1} (\check{x}_j^k)^2 R_{ii}^k = \sum_{k=1}^m \mathrm{Tr}(R^k)\|\check{\mathbf{x}}^k\|^2 \leq \max_k \|\check{\mathbf{x}}^k\|^2
$$

Denote $max := \arg\max_k \|\check{\mathbf{x}}^k\|^2$. By the last inequality there is $\check{W} \in \check{\mathcal{L}}$ such that

$$
\sum_{i=1}^m (\check{\mathbf{x}}^i)^\top (W - \check{W})^\top R^i (W - \check{W})\check{\mathbf{x}}^i \leq \varepsilon_1^2 \|\check{\mathbf{x}}^{max}\|^2
$$

We have

$$
\begin{aligned}
\|W\mathbf{x} - \check{W}\check{\mathbf{x}}\|_R &\leq \|\mathbf{x} - \check{\mathbf{x}}\|_{W^\top RW} + \varepsilon_1\|\check{\mathbf{x}}^{max}\| \\
&\leq \|\mathbf{x} - \check{\mathbf{x}}\|_{W^\top RW} + \varepsilon_1(\|\check{\mathbf{x}}^{max} - \mathbf{x}^{max}\| + \|\mathbf{x}^{max}\|) \\
&\leq \|\mathbf{x} - \check{\mathbf{x}}\|_{W^\top RW} + \|\check{\mathbf{x}}^{max} - \mathbf{x}^{max}\|_{\varepsilon_1^2 I} + \varepsilon_1 M \\
&\leq \sqrt{2}\sqrt{\|\mathbf{x} - \check{\mathbf{x}}\|_{W^\top RW}^2 + \|\check{\mathbf{x}}^{max} - \mathbf{x}^{max}\|_{\varepsilon_1^2 I}^2} + \varepsilon_1 M \\
&\leq \sqrt{2}\sqrt{r^2\varepsilon_2^2 + d_1\varepsilon_1^2\varepsilon_2^2} + \varepsilon_1 M
\end{aligned}
$$

$\square$

$\square$

*Proof.* (of Lemma 3.3) Let $\check{\mathcal{X}}$ be an $\varepsilon$-multicover of $\mathcal{X}$ of size $M(\mathcal{X}, \varepsilon)$. By lemma 3.1 for any $\mathbf{x} \in \mathcal{X}$ there is a distribution $\mathcal{D}_\mathbf{x}$ on $\check{\mathcal{X}}$ such that if $X \sim \mathcal{D}_\mathbf{x}$ then $X$ is an $\varepsilon$-estimator of $\mathbf{x}$. In particular, for any coordinate $i \in [d]$ and $j \in [m]$ we have

$$
\mathbb{E}_X(X_i^j - x_i^j)^2 = \mathbb{E}_X(X^j - \mathbf{x}^j)^\top E_{ii}(X^j - \mathbf{x}^j) \leq \varepsilon^2
$$

Denote $k = \left\lceil \log_{r/2}(d) \right\rceil$. By the above equation and lemma 2.1 we conclude that if $X(1), \ldots, X(k) \sim \mathcal{D}_\mathbf{x}$ then for every $i \in [d]$ and $j \in [m]$

$$
\Pr\left(\exists i \in [d] \text{ s.t. } |\mathrm{median}(X(1)_i^j, \ldots, X(k)_i^j) - x_i^j| > r\varepsilon\right) < d\left(\frac{2}{r}\right)^k \leq 1
$$

in particular, there exists $\mathbf{x}(1), \ldots, \mathbf{x}(k) \in \check{\mathcal{X}}$ such that for any $i \in [d]$ and $j \in [m]$, $|\mathrm{median}(x(1)_i^j, \ldots, x(k)_i^j) - x_i^j| \leq r\varepsilon$. This implies that $\mathrm{median}(\check{\mathcal{X}}^k)$ is an $\varepsilon$-cover of $\mathcal{X}$ w.r.t. the $\ell^\infty$ norm. This concludes the proof as $|\mathrm{median}(\check{\mathcal{X}}^k)| \leq |\check{\mathcal{X}}|^k$ $\square$

### A.4 STRONGLY BOUNDED ACTIVATION FUNCTIONS

**Lemma A.6** ($\beta$-Swish Activation Ramachandran et al. (2017))**.** *For a constant $\beta \geq 0$, the function $\frac{x}{1 + e^{-\beta x}}$ is strongly-bounded*

It is shown in Daniely & Granot (2019) that

**Fact A.7.** *If $\rho$ is $B$-strongly-bounded then $\rho$ is analytic and its Taylor coefficients around any point are bounded by $B^n$ for any $n \geq 1$.*

*Proof.* For the case of $\beta = 0$ the swish becomes a linear function, and the claim is trivial. For $\beta > 1$, consider the complex function $f(z) = \frac{z}{1+e^{-\beta z}}$. It is defined in the strip $\{z = x + iy : |y| < \frac{1}{\beta}\pi\}$. By Cauchy integral formula, for any $r < \frac{\pi}{\beta}$, $a \in \mathbb{R}$ and $n \geq 0$,

$$f^{(n)}(a) = \frac{n!}{2\pi i} \int_{|z-a|=r} \frac{f(z)}{(z-a)^{n+1}}$$

It follows that

$$\left|f^{(n)}(a)\right| \leq \frac{n!}{r^n} \max_{|z-a|=r} |f(z)| \leq \frac{n!}{r^n} \max_{x+iy:|y|<r} |f(x+iy)|$$

Now, if $|y| < r < \frac{\pi}{2\beta}$, we have

$$|f(x+iy)| = \frac{|x+iy|}{|1 + e^{-i\beta y}e^{-\beta x}|} \leq \frac{r}{|1 + \cos(-\beta y)e^{-\beta x}|} \leq \frac{r}{|1 + \cos(\beta r)e^{-\beta x}|} \leq r$$

This implies that $\frac{x}{1+e^{-\beta x}}$ is strongly bounded. $\square$

**Lemma A.8** (Hyperbolic Tangent). *The function $\frac{e^{2x}-1}{e^{2x}+1}$ is strongly-bounded*

*Proof.* Consider the complex function $f(z) = \frac{e^{2z}-1}{e^{2z}+1}$. It is defined in the strip $\{z = x + iy : |y| < \frac{1}{2}\pi\}$. By Cauchy integral formula, for any $r < \frac{\pi}{2}\pi$, $a \in \mathbb{R}$ and $n \geq 0$,

$$f^{(n)}(a) = \frac{n!}{2\pi i} \int_{|z-a|=r} \frac{f(z)}{(z-a)^{n+1}}$$

It follows that

$$\left|f^{(n)}(a)\right| \leq \frac{n!}{r^n} \max_{|z-a|=r} |f(z)| \leq \frac{n!}{r^n} \max_{x+iy:|y|<r} |f(x+iy)|$$

Now, if $|y| < r < \frac{\pi}{8}$ we have that

$$|f(x+iy)| = \frac{|e^{2x}e^{2iy} - 1|}{|e^{2x}e^{2iy} + 1|} \leq \frac{2\max\{e^{2x}, 1\}}{|e^{2x}e^{2iy} + 1|} \leq \frac{2\max\{e^{2x}, 1\}}{|e^{2x}\cos(2y) + 1|} \leq \frac{2\max\{e^{2x}, 1\}}{|e^{2x}\cos(2r) + 1|} \leq \frac{2}{\cos(2r)}$$

This implies that $\frac{e^{2x}-1}{e^{2x}+1}$ is strongly bounded. $\square$

**Lemma A.9** (3.8). *Let $p(x) = \sum_{i=0}^{k} a_i X^i$ be a polynomial with $|a_i| \leq B^i$ and suppose that $\mathcal{X} \subset \left[-\frac{1}{8B}, \frac{1}{8B}\right]^{d,m}$. Then, for any Let $0 < \varepsilon \leq 1$,*

$$M(p(\mathcal{X}), \varepsilon) \leq \left(M\left(\mathcal{X}, \frac{\varepsilon}{8B}\right)\right)^{\frac{k(k+1)}{2}}$$

*Proof.* As $M(p(\mathcal{X}), \varepsilon) = M(p(\mathcal{X}) - a_0, \varepsilon)$ we can assume w.l.o.g. that $a_0 = 0$. Denote $a = \frac{1}{8B}$ and $\varepsilon_i = i2^{-2i-1}\varepsilon$. Note that since for $-1 < x < 1$, $\frac{1}{(1-x)^2} = \sum_{i=1}^{\infty} ix^{i-1}$ we have that

$$\sum_{i=1}^{k} \varepsilon_i \leq \frac{\varepsilon}{4} \sum_{i=1}^{\infty} i(1/2)^{i-1} = \frac{\varepsilon}{4} \frac{1}{(1/2)^2} = \varepsilon \tag{4}$$

Hence, we have that

$$
\begin{aligned}
M\left(p(\mathcal{X}),\varepsilon\right) \quad &\overset{p(\mathcal{X})\subseteq\sum_{i=1}^{k}a_i\mathcal{X}^i}{\leq} \quad M\left(\sum_{i=1}^{k}a_i\mathcal{X}^i,\varepsilon\right) \\
&\overset{\sum_{i=1}^{k}\varepsilon_i\leq\varepsilon}{\leq} \quad M\left(\sum_{i=1}^{k}a_i\mathcal{X}^i,\sum_{i=1}^{k}\varepsilon_i\right) \\
&\overset{Lem.3.2}{\leq} \quad \prod_{i=1}^{k}M\left(a_i\mathcal{X}^i,\varepsilon_i\right) \\
&\overset{Lem.3.2}{\leq} \quad \prod_{i=1}^{k}M\left(\mathcal{X}^i,\varepsilon_i/|a_i|\right) \\
&\overset{|a_i|\leq B^i}{\leq} \quad \prod_{i=1}^{k}M\left(\mathcal{X}^i,\varepsilon_i/B^i\right) \\
&\overset{Claim3}{\leq} \quad \prod_{i=1}^{k}M\left(\mathcal{X}^i,\left(a+\frac{\varepsilon_i}{i(2a)^{i-1}B^i}\right)^i-a^i\right) \\
&\overset{Lem.3.2}{\leq} \quad \prod_{i=1}^{k}\left(M\left(\mathcal{X},\frac{\varepsilon_i}{i(2a)^{i-1}B^i}\right)\right)^i \\
&= \quad \prod_{i=1}^{k}\left(M\left(\mathcal{X},\frac{i2^{-2i-1}\varepsilon}{i(2a)^{i-1}B^i}\right)\right)^i \\
&= \quad \prod_{i=1}^{k}\left(M\left(\mathcal{X},\frac{\varepsilon}{(8aB)^{i-1}8B}\right)\right)^i \\
&\overset{8aB=1}{=} \quad \prod_{i=1}^{k}\left(M\left(\mathcal{X},\frac{\varepsilon}{8B}\right)\right)^i \\
&= \quad \left(M\left(\mathcal{X},\frac{\varepsilon}{8B}\right)\right)^{\frac{k(k+1)}{2}}
\end{aligned}
$$

**Claim 3.** $\left(a+\frac{\varepsilon_i}{i(2a)^{i-1}B^i}\right)^i-a^i\leq\frac{\varepsilon_i}{B^i}$

*Proof.* Denote $f(x)=x^i$. Since $f$ is convex on $\mathbb{R}_+$ we have

$$
f\left(a+\frac{\varepsilon_i}{i(2a)^{i-1}B^i}\right)-f(a)\leq f'\left(a+\frac{\varepsilon_i}{i(2a)^{i-1}B^i}\right)\frac{\varepsilon_i}{i(2a)^{i-1}B^i}
$$

Now, $\frac{\varepsilon_i}{i(2a)^{i-1}B^i}\leq a \Leftrightarrow \varepsilon_i\leq i2^{i-1}(aB)^i \Leftrightarrow i2^{-2i-1}\varepsilon\leq i2^{i-1}(aB)^i \Leftrightarrow \varepsilon\leq 8^i(aB)^i=1$. Hence $\frac{\varepsilon_i}{i(2a)^{i-1}B^i}\leq a$. Since $f'$ is monotone on $\mathbb{R}_+$ we have

$$
f\left(a+\frac{\varepsilon_i}{i(2a)^{i-1}B^i}\right)-f(a)\leq f'(2a)\frac{\varepsilon_i}{i(2a)^{i-1}B^i}
$$

This translate to

$$
\left(a+\frac{\varepsilon_i}{i(2a)^{i-1}B^i}\right)^i-a^i\leq i(2a)^{i-1}\frac{\varepsilon_i}{i(2a)^{i-1}B^i}=\frac{\varepsilon_i}{B^i}
$$

$\square$

$\square$

## A.5 BOUNDING THE MULTICOVERING NUMBER OF NEURAL NETWORKS

*Proof of 3.13.* As $M(\mathcal{H}, m, \epsilon)$ is monotonically decreasing with $\epsilon$, and the inequality is up to constant, it is enough to prove the lemma for $\epsilon \leq \frac{1}{32B}$. Denote

$$\mathcal{L}_i = \{W : \|W - W_i^0\| \leq r, \ \|W - W_i^0\|_F \leq R\}$$

Fix examples $\mathbf{x}^1, \ldots, \mathbf{x}^m \in B_{\sqrt{d}}^d$. Denote $\mathcal{X}_0 = \{(\mathbf{x}^1, \ldots, \mathbf{x}^m)\} \subset \mathbb{R}^{d,m}$. For $1 \leq i \leq t$ denote $\mathcal{X}_i = \rho(\mathcal{L}_i \mathcal{X}_{i-1})$. We need to show that

$$M(\mathcal{X}_t, \epsilon) \lesssim (\log(dm) + \log^2(d/\epsilon))^t \log(dR) \frac{dR^2}{\epsilon^2}$$

where the hidden constant does not depend on the choice of $\mathbf{x}^1, \ldots, \mathbf{x}^m$.

Note that $\mathcal{X}_i \subset (B_M^d)^m$ for $M \lesssim \sqrt{D}$. Let $k = \left\lceil \log_8(\sqrt{d}/\epsilon) \right\rceil$ and choose $\epsilon_2 > 0$ such that for $\epsilon_1 = \frac{\epsilon_2}{\sqrt{d}+M}$ we have

$$\epsilon = 8B\epsilon_2\sqrt{2r^2 + 2\epsilon_1^2 d} + 8B\epsilon_1 M + \sqrt{d}8^{-(k+1)}$$

Note that

$$\epsilon \leq 8B\epsilon_2\sqrt{2r^2 + 2} + 8B\epsilon_2 + \epsilon/8 \Rightarrow \epsilon \leq \frac{8}{7}8B(\sqrt{2r^2 + 2} + 2)\epsilon_2 =: C\epsilon_2$$

We have

$$
\begin{aligned}
M(\mathcal{X}_t, \epsilon) &= M(\rho(\mathcal{L}_t \mathcal{X}_{t-1}), \epsilon) \\
&\overset{\text{Lemma 3.9}}{\leq} \left(M\left(\mathcal{L}_t \mathcal{X}_{t-1}, \frac{1}{32B}\right)\right)^{\lceil \log_2(dm) \rceil} \left(M\left(\mathcal{L}_t \mathcal{X}_{t-1}, \epsilon_2\sqrt{2r^2 + 2\epsilon_1^2 d} + \epsilon_1 M\right)\right)^{\frac{k(k+1)}{2}} \\
&\leq \left(M\left(\mathcal{L}_t \mathcal{X}_{t-1}, \epsilon_2\sqrt{2r^2 + 2\epsilon_1^2 d} + \epsilon_1 M\right)\right)^{\lceil \log_2(dm) \rceil + \frac{k(k+1)}{2}} \\
&\overset{\text{Lemma 3.2}}{\leq} (M(\mathcal{L}_t, \epsilon_1) M(\mathcal{X}_{t-1}, \epsilon_2))^{\lceil \log_2(dm) \rceil + \frac{k(k+1)}{2}} \\
&\overset{\epsilon/C \leq \epsilon_2}{\leq} (M(\mathcal{L}_t, \epsilon_1) M(\mathcal{X}_{t-1}, \epsilon/C))^{\lceil \log_2(dm) \rceil + \frac{k(k+1)}{2}}
\end{aligned}
$$

By lemma 2.5 and since $\epsilon_1 = \frac{\epsilon_2}{\sqrt{d}+M}$ we have

$$\log(M(\mathcal{L}_t, \epsilon_1)) \leq \left\lceil 2(d + M^2)\frac{2R^2 + 1/4}{\epsilon_2^2} \right\rceil \log(4d^4 \lceil R \rceil + 6d^2)$$

Thus we get

$$
\begin{aligned}
\log(M(\mathcal{X}_t, \epsilon)) &= \log(M(\rho(\mathcal{L}_t \mathcal{X}_{t-1}), \epsilon)) \\
&\lesssim (\log(dm) + \log^2(d/\epsilon))\left(\frac{dR^2}{\epsilon^2}\log(dR) + M(\mathcal{X}_{t-1}, \epsilon/C)\right)
\end{aligned}
$$

Inductively, we get that

$$\log(M(\mathcal{X}_t, \epsilon)) \lesssim (\log(dm) + \log^2(d/\epsilon))^i \log(dR)\frac{dR^2}{\epsilon^2}$$

$\square$

## A.6 BOUNDING REPRESENTATIVENESS WITH MULTICOVER

**Lemma A.10** (3.11). *Let $\ell : \mathbb{R}^d \times \mathcal{Y} \to \mathbb{R}$ be L-Lipschitz w.r.t. $\|\cdot\|_\infty$ and B-bounded. Assume that for any $\frac{\sqrt{n}B}{\sqrt{m}8L} \leq \varepsilon \leq 1$, $\ln M(\mathcal{H}, m, \varepsilon) \leq \frac{n}{\varepsilon^2}$. Then for any distribution $\mathcal{D}$ on $\mathcal{Z}$*

$$\mathbb{E}_{S \sim \mathcal{D}^m} \mathrm{rep}_\mathcal{D}(S, \mathcal{H}) \lesssim \frac{(L + B)\sqrt{n}}{\sqrt{m}}\sqrt{\log(dm)}\log(m)$$

*Proof.* First note that by Lemma 3.3,

$$N_2(\ell \circ \mathcal{H}, m, \varepsilon) \leq N_\infty(\mathcal{H}, m, \varepsilon/L) \leq \cdot M(\mathcal{H}, m, \varepsilon/(4L))^{\lceil \log_2(dm) \rceil}$$

Denote

$$A = B2^{-M+1} + \frac{12B}{\sqrt{m}} \sum_{k=1}^{M} 2^{-k} \sqrt{\ln\left(N_2(\ell \circ \mathcal{H}, m, B2^{-k})\right)}$$

We have

$$
\begin{aligned}
A &\leq B2^{-M+1} + \frac{12B\sqrt{\lceil \log_2(dm) \rceil}}{\sqrt{m}} \sum_{k=1}^{M} 2^{-k} \sqrt{\ln\left(M(\mathcal{H}, m, B2^{-k}/(4L))\right)} \\
&\leq B2^{-M+1} + \frac{12B\sqrt{n\lceil \log_2(dm) \rceil}}{\sqrt{m}} \sum_{k=1}^{M} 2^{-k} \sqrt{\left(\frac{2^{2k}16L^2}{B^2} + 1\right)} \\
&\leq B2^{-M+1} + \frac{12B\sqrt{n\lceil \log_2(dm) \rceil}}{\sqrt{m}} \sum_{k=1}^{M} \sqrt{\left(\frac{16L^2}{B^2} + 2^{-2k}\right)} \\
&\leq B2^{-M+1} + \frac{12B\sqrt{n\lceil \log_2(dm) \rceil}}{\sqrt{m}} \sum_{k=1}^{M} \left(\frac{4L}{B} + 2^{-k}\right) \\
&\leq B2^{-M+1} + \frac{12B\sqrt{n\lceil \log_2(dm) \rceil}}{\sqrt{m}} \left(\frac{4LM}{B} + 1\right)
\end{aligned}
$$

Choosing $M = \lceil \log_2\left(\sqrt{\frac{m}{n}}\right) \rceil$ we get,

$$A \leq B\sqrt{\frac{n}{m}} + \frac{12B\sqrt{n\lceil \log_2(dm) \rceil}}{\sqrt{m}} \left(\frac{4L\log(m)}{B} + 1\right)$$

$\square$

# B   APPROXIMATE DESCRIPTION LENGTH AND MULTICOVER

In this section we show that multicover is closely related to the notion of approximate description length (ADL) as defined by Daniely & Granot (2019). We start with a definition that is slightly different from the definition used in Daniely & Granot (2019). We say that $\mathcal{X}$ has $\varepsilon$-*ADL* of $n$ if there is a protocol between two entities, Alice and Bob with the following properties. Upon seeing $\mathbf{x} \in \mathcal{X}$, Alice, that is allowed to use randomness, sends a message $s \in \{0,1\}^n$ to Bob. Upon seeing $s$, Bob generates a vector $\hat{\mathbf{x}}$ that is an $\varepsilon$-estimator to $\mathbf{x}$. Formally, there is a probability space $(\Omega, P)$ (representing Alice's randomness) and functions $A : \mathcal{X} \times \Omega \to \{0,1\}^n$ and $B : \{0,1\}^n \to \mathbb{R}^{d,m}$ such that for any $\mathbf{x} \in \mathcal{X}$ the random variable $\omega \mapsto B(A(\mathbf{x}, \omega))$ is an $\varepsilon$-estimator of $\mathbf{x}$. We denote by $\mathrm{ADL}(\mathcal{X}, \varepsilon)$ the minimal $k$ for which $\mathcal{X}$ has an $\varepsilon$-ADL of $k$.

The following lemma shows that ADL is closely related to multicover.

**Lemma B.1.** $\mathrm{ADL}(\mathcal{X}, \varepsilon) = \lfloor \log_2\left(M(\mathcal{X}, \varepsilon)\right) \rfloor$

*Proof.* Observe that $\mathcal{X}$ has $\varepsilon$-ADL of $n$ if and only if there is a set $\check{\mathcal{X}}$ such that for any $\mathbf{x} \in \mathcal{X}$ there is a random vector $X \in \check{\mathcal{X}}$ that is a $\varepsilon$-estimator of $\mathbf{x}$. By lemma 3.1 this is valid if and only if $\check{\mathcal{X}}$ is an $\varepsilon$-multicover. It follows that $\mathrm{ADL}(\mathcal{X}, \varepsilon)$ is the minimal $k$ for which $\mathcal{X}$ has an $\varepsilon$-multicover of size $2^k$. In other words, $\mathrm{ADL}(\mathcal{X}, \varepsilon) = \lfloor \log_2\left(M(\mathcal{X}, \varepsilon)\right) \rfloor$. $\square$

We next turn to the definition used in Daniely & Granot (2019). We define *unbiased $\varepsilon$-ADL*, by making two modification to the definition of $\varepsilon$-ADL. First, we require that $\mathbb{E}_{\omega \sim P} B(A(\mathbf{x}, \omega)) = \mathbf{x}$. Second, we allow sending messages of unbounded length (i.e. a message in $\{0,1\}^*$), and just require that the expected number of sent bits will be at most $n$. We denote by $u\mathrm{ADL}(\mathcal{X}, \varepsilon)$ the minimal $k$ for which $\mathcal{X}$ has an unbiased $\varepsilon$-ADL of $k$. We note that Daniely & Granot (2019) defined the ADL of $\mathcal{X}$ to be $u\mathrm{ADL}(\mathcal{X}, 1)$. The following lemma connects unbiased ADL and ADL by showing that ignoring poly-logarithmic factors, $u\mathrm{ADL}(\mathcal{X}, 1) \leq k$ if and only if $\mathrm{ADL}(\mathcal{X}, \varepsilon) \leq \frac{k}{\varepsilon^2}$. By lemma B.1 this happens if and only if $\log_2(M(\mathcal{X}, \varepsilon)) \leq \frac{k}{\varepsilon^2}$.

**Lemma B.2.** *Fix* $\mathcal{X} \subset \mathbb{R}^{d,m}$. *We have*

- $\forall 0 < \epsilon \leq 1,\ \mathrm{ADL}(\mathcal{X}, \varepsilon) \leq O\left(\frac{u\mathrm{ADL}(\mathcal{X},1)}{\varepsilon^2}\right)$

- *If* $\mathrm{ADL}(\mathcal{X}, \varepsilon) \leq \frac{k}{\varepsilon^2}$ *for any* $0 < \epsilon \leq 1$ *then* $u\mathrm{ADL}(\mathcal{X}, 1) = O\left(\log^2(dm)k\right)$

*Where the constant in the big-O notation are universal.*

*Proof.* (sketch) Denote $k = u\mathrm{ADL}(\mathcal{X}, 1)$. Given $\mathbf{x} \in \mathcal{X}$ and using $O\left(\frac{k}{\epsilon^2}\right)$ expected bits Alice can send to Bob $\left\lceil \frac{1}{\epsilon^2} \right\rceil$ independent and unbiased 1-estimators of $\mathbf{x}$. If Bob averages these estimators, he gets an $\epsilon$-estimator of $\mathbf{x}$. This implies that $u\mathrm{ADL}(\mathcal{X}, \varepsilon) \leq O\left(\frac{u\mathrm{ADL}(\mathcal{X},1)}{\varepsilon^2}\right)$. It is therefore enough to show that $\mathrm{ADL}(\mathcal{X}, \varepsilon) \leq O\left(u\mathrm{ADL}(\mathcal{X}, \varepsilon/\sqrt{2})\right)$. By Lemma B.1 it is enough to show that $\log(M(\mathcal{X}, \varepsilon)) \leq O\left(u\mathrm{ADL}(\mathcal{X}, \varepsilon/\sqrt{2})\right)$.

Denote $k = u\mathrm{ADL}(\mathcal{X}, \varepsilon/\sqrt{2})$ and fix a probability space $(\Omega, P)$ and functions $A : \mathcal{X} \times \Omega \to \{0,1\}^*$ and $B : \{0,1\}^* \to \mathbb{R}^{d,m}$ such that for any $\mathbf{x} \in \mathcal{X}$ the random variable $\omega \mapsto B(A(\mathbf{x}, \omega))$ is an unbiased $(\varepsilon/\sqrt{2})$-estimator of $\mathbf{x}$, and $\mathbb{E}_\omega \mathrm{len}(A(\mathbf{x}, \omega)) \leq k$. Fix $R \in \mathcal{R}_1^{d,m}$. We have $\mathbb{E}_\omega \|\mathbf{x} - B(A(\mathbf{x}, \omega))\|_R^2 \leq \epsilon^2/2$ By Markov inequality, there exists $\omega$ such that $\|\mathbf{x} - B(A(\mathbf{x}, \omega))\|_R^2 \leq \epsilon^2$ and $\mathrm{len}(A(\mathbf{x}, \omega)) \leq 2k$. This implies that $\check{\mathcal{X}} := \{B(s) : s \in \{0,1\}^*,\ \mathrm{len}(s) \leq 2k\}$ is a $\epsilon$-cover of $\mathcal{X}$ w.r.t. $R$. This is true for any $R \in \mathcal{R}_1^{d,m}$, and therefore $\check{\mathcal{X}}$ is a $\epsilon$-multicover of $\mathcal{X}$. This implies that $\log(M(\mathcal{X}, \varepsilon)) \leq 4k$

For the second item, let $X_n$ and $\bar{X}_n$ be $\frac{\epsilon}{\sqrt{2^n}}$-estimators of $\mathbf{x}$, which can be encoded using $\frac{k2^n}{\epsilon^2}$ bits each. Let $Z_n$ be a r.v. that is $2^n$ w.p. $2^{-n}$ and 0 otherwise. Assume that all these random variables are independent. Consider now the estimator

$$X^N = X_1 + \sum_{n=1}^N Z_n(X_{n+1} - \bar{X}_n)$$

We first claim that Bob can generate such an estimator using $O\left(\frac{kN}{\epsilon^2}\right)$ expected bits set from Alice. Indeed, Alice can first sample the $Z_n$'s. Then, for any $n$, if $Z_n \neq 0$, send the index $n$ using $O(\log(n))$ bits as well as $X_n$ and $\bar{X}_n$ using $\frac{k2^n}{\epsilon^2}$. The expected number of sent bits is $O\left(2^{-n}\frac{k2^n}{\epsilon^2}\right) = O\left(\frac{k}{\epsilon^2}\right)$. The total expected number of sent bits is therefore $O\left(\frac{kN}{\epsilon^2}\right)$ bits. We next show that $X^N$ is an $\left(\epsilon\sqrt{1+4N}\right)$-estimator of $\mathbf{x}$. Indeed, for any unit vector $\mathbf{u}$ we have

$$
\begin{aligned}
\mathrm{Var}\left(\langle \mathbf{u}, X^N \rangle\right) &= \mathrm{Var}(\langle \mathbf{u}, X_1 \rangle) + \sum_{n=1}^N \mathrm{Var}\left(Z_n \langle \mathbf{u}, X_{n+1} - X_n \rangle\right) \\
&\leq \epsilon^2 + \sum_{n=1}^N \mathbb{E}\left(Z_n \langle \mathbf{u}, X_{n+1} - X_n \rangle\right)^2 \\
&= \epsilon^2 + \sum_{n=1}^N \mathbb{E}Z_n^2 \mathbb{E}\left(\langle \mathbf{u}, X_{n+1} - X_n \rangle\right)^2 \\
&= \epsilon^2 + \sum_{n=1}^N 2^n \mathbb{E}\langle \mathbf{u}, X_{n+1} - X_n \rangle^2 \\
&\leq \epsilon^2 + 2\sum_{n=1}^N 2^n \left(\mathbb{E}\langle \mathbf{u}, X_{n+1} - \mathbf{x} \rangle^2 + \mathbb{E}\langle \mathbf{u}, X_n - \mathbf{x} \rangle^2\right) \\
&\leq \epsilon^2 + \epsilon^2 2\sum_{n=1}^N 2^n \left(2^{-(n+1)} + 2^{-n}\right) \\
&\leq \epsilon^2(1 + 4N)
\end{aligned}
$$

Let $Y^N$ be an unbiased $(1/2)$-estimator of $\mathbf{y} := \mathbf{x} - \mathbb{E}X^N = \mathbf{x} - EX_{N+1}$. Since $X_{N+1}$ is $\left(\frac{\epsilon}{2^{(N+1)/2}}\right)$-estimator of $\mathbf{x}$, we have that the absolute value of each coordinate of $\mathbf{y}$ is at most

$\frac{\epsilon}{2^{(N+1)/2}}$. Thus, Alice can send $\frac{1}{2}$ estimator of $\mathbf{y}$ as follows: for any $i \in [m]$ and $j \in [d]$, w.p. $|y_j^i|$ send $\mathrm{sign}(y_j^i)$ and the indices $i$ and $j$. If the pair $(i,j)$ were sent, Bob will define $Y_j^i = \mathrm{sign}(y_j^i)$. Otherwise, he will define $Y_j^i = 0$. It is not hard to verify (see Daniely & Granot (2019) for details) that $Y$ is an unbiased $\frac{\epsilon}{2^{(N+1)/2}}$-estimator of $Y$. Likewise, the expected number of sent bits per coordinate is $O\left(\frac{\epsilon \log(md)}{2^{(N+1)/2}}\right)$, resulting with a total cost of $O\left(\frac{\epsilon md \log(md)}{2^{(N+1)/2}}\right) = O\left(\frac{md \log(md)}{2^{(N+1)/2}}\right)$ bits.

Finally, $X = X^N + Y^N$ is an unbiased $\sqrt{1/2 + \epsilon^2(1+4N)}$-estimator of $\mathbf{x}$ which costs $O\left(\frac{md \log(md)}{2^{(N+1)/2}} + \frac{kN}{\epsilon^2}\right)$ bits to encode. Choosing $\epsilon = \sqrt{1/(2+4N)}$ and $N = 2\log_2(md)$ we gen an unbiased 1-estimator of $\mathbf{x}$ which costs $O\left(k \log^2(md)\right)$ bits to encode.

$\square$

