# OpenReview forum: "A Multicover Approach to Neural Networks Sample Complexity"
_ICLR.cc/2025/Conference — Submitted to ICLR 2025_

### Official Review · Reviewer_UGv1 · 2024-10-21

**Soundness:** 1
**Presentation:** 1
**Contribution:** 1
**Rating:** 1
**Confidence:** 5

**Summary:**

The manuscript tries to derive multi-covering number bounds to recover a recent result of Daniely & Granot (2019). To this end, the paper introduces the concepts of generalized multi-covering, related facts, and lemmas of covering numbers and packing numbers and tres to derive results on generalized multi-covering.

**Strengths:**

The manuscripts try to derive the parallel results for the defined multi-covering number following those of the covering number.

**Weaknesses:**

A very poorly written manuscript. There is no readability. The manuscript is just a listing of basic facts and simple existing lemmas of covering numbers with lots of typos and unclear definitions. The final claimed result is nothing but a recovery of the results in Daniely & Granot (2019) without proper comparison with existing results and shows no advantages over the existing one (overclaimed in the abstract).

**Questions:**

The manuscript is full of unclear and unreasonable statements, just list one here:

 -  In abstract: ``Alas, standard techniques for bounding covering numbers fail in estimating the covering numbers of many classes of neural networks. We introduce a generalization of covers, called multicovers, which are covered w.r.t. many metrics simultaneously. Contrary to standard covering numbers, multicovering numbers behave better with the layer-wise structure in neural networks"

Standard techniques for bounding covering numbers have successfully derived the covering numbers of many classes of neural networks[1][2][3][4], why did it fail? I did not see any literature review of the existing works on covering numbers in the ``introduction" and other parts of the paper. There are in total 11 references in the manuscripts, did the author really properly know the basic advances for the topic before they drafted the paper?


[1] Neyshabur, B., Bhojanapalli, S., and Srebro, N. A
pac-bayesian approach to spectrally-normalized margin bounds for neural networks. arXiv preprint
arXiv:1707.09564, 2017.
[2] Bartlett, P. L., Foster, D. J., and Telgarsky, M. J. Spectrallynormalized margin bounds for neural networks. Advances
in neural information processing systems, 30, 2017.
[3] Wei, Colin, and Tengyu Ma. Data-dependent sample complexity of deep neural networks via lipschitz augmentation. Advances in Neural Information Processing Systems 32 (2019).
[4] Lin, S. and Zhang, J. Generalization bounds for convolutional neural networks. arXiv preprint arXiv:1910.01487,
2019.

---

### Official Review · Reviewer_A5dk · 2024-10-21

**Soundness:** 3
**Presentation:** 2
**Contribution:** 2
**Rating:** 3
**Confidence:** 3

**Summary:**

This paper introduces a new notion of multicover to upper bound the sample complexity of certain classes of neural networks. While traditional covering number approaches are not well-suited to inductive reasoning, the new multicover calculus allows for such inductive reasoning. This provides a new set of tools to analyze the sample complexity of deep neural networks of arbitrary depth, by reasoning inductively on the layers. In particular, the authors use their new multicover calculus to recover known sample complexity bounds for neural networks with bounded Froebenius norm. Additionally, it is shown that the introduced multi cover calculus is closely related to the notion of Approximate Description Length.

**Strengths:**

- the possibility to work inductively on multi-layer models may prove to be a useful tools to derive new sample complexity bounds.

 - The authors provide a lot of details regarding the multicover calculus, ie, the properties of the newly introduced multicovers. This may be used as a toolbox for other research on sample complexity.

- The introduced methods allows to recover known smample complexity bounds.

**Weaknesses:**

- The new multicover technique is only used to recover known bounds. Therefore, it is not clear that the new techniques will be useful to prove new results. The authors claim that the fact that covering number do not work well inductively is a major flaw, however solving this major flaw does not seem to lead to better theoretical results directly. It could be beneficial to include additional discussion on the type of settings/assumptions/architectures under which the multicovers can improve existing results or prove new ones.

 - Several `well-known' results are cited without references, including these references would help the readers a lot, for instance in Lines 72, 78. In general a more in-depth literature review (regarding sample complexity bounds neural networks and how they are related to coverings) might be missing (there are only 11 references, including classical textbooks). For me, it makes it hard to judge the novelty of the work. I am not an expert on the question, but the following papers might be relevant: Philip M Long and Hanie Sedghi. Generalization bounds for deep convolutional neural networks. Simon Du and Jason Lee. On the power of over-parametrization in neural networks with quadratic activation. Guohao Shen - Exploring the Complexity of Deep Neural Networks through Functional Equivalence.



 - Several notations are used before they are defined, eg, in Section 1.1 or Lemma 2.1.  The notations section should be moved at the beginning of Section 1.1.

 - As multicover calculus seems to be very closely related to the previously introduced notion of Approximate Description Length (ADL). Therefore, one may ask what do multi covers bring compared to ADL. This fact should be discussed in greater detail in the main part of the paper.

**Questions:**

- According to Lemma B.1, the multicovers seem to be very similar to the notion of ADL. Therefore, one may wonder what is the interest of introducing the multicovers if ADL is already available. Additionally, does ADL allow for inductive reasoning over the depth of the networks? What can the multicovers approach bring when compared to ADL?

 - Why is the lack of inductive reasnoning a major flaw of existing approaches? Is it obvious that we will be able to prove better sample complexity bounds using inductive reasoning?

 - Does you multicover technique apply to convolutional architectures?

**Other minor remarks and typos**

 - In the introduction, there are a lot of words like often, nearly, potentially,... It could enhance the writing to replace them with more precise statements.

  - Line 56: Radamacher $\to$ Rademacher. nevertheless $\to$ Nevertheless. proved $\to$ proven.

  - The notations section (Section 2.1) appears on page 3, but a lot of these notations are already used on page 2 and were not introduced before.

  - Line 72/73: do you have a reference for this result?

  - Line 78: what is $\mathcal{X}_2$, if I am correct it has not been defined at this point.

  - Line 92: chows $\to$ shows

  - Table 1 is impossible to read. You should find another way to compare with the literature.

  - Line 144: with that that $\to$ that

  - Line 371: the end of the sentence is missing

---

### Official Review · Reviewer_LVMB · 2024-11-01

**Soundness:** 2
**Presentation:** 1
**Contribution:** 3
**Rating:** 5
**Confidence:** 1

**Summary:**

The authors propose a generalisation of the notion of covering numbers to better support the "compositional" nature of neural networks.
They use this tool to improve the sample complexity bounds for neural networks.

**Strengths:**

The understanding the generalisation capabilities of artificial neural networks is a key challenge in the field.

**Weaknesses:**

The authors show little care for presentation.
There are many typos, concepts being used before their definition, and, more generally, the writing makes ideas much harder to understand than necessary.
It gives the impression that the work was produced in a hurry.

Still, to the extent I managed to check, what the authors write seems to be correct in essence, at least, so I guess this community tolerates the disregard for communication.
Accordingly, I concluded that the fairest course of action would be
1. Lean towards rejection based on presentation, as this is the only aspect I managed to evaluate properly;
2. To assign minimal confidence to the review (suitably warning the AC), as it is unfeasible for me to fully evaluate this work. This also mitigates the weight of the rejection recommendation.

To be clear, I normally would have enough expertise in the relevant areas to review the paper so the allocation system is not to blame.
It is the presentation that, in my opinion, makes it so that reviewing this work with a reasonable amount of effort requires someone specialized in the very specific literature around it (e.g., authors of related works).

**Questions:**

None.

---

### Official Review · Reviewer_9ZJi · 2024-11-07

**Soundness:** 3
**Presentation:** 2
**Contribution:** 2
**Rating:** 5
**Confidence:** 3

**Summary:**

The paper introduces a novel method known as multicovers, which allows for estimating covering numbers for deep neural networks by considering multiple metrics simultaneously rather than relying on a single metric. This approach behaves better than the standard covering number for the networks' layer-wise structure. The authors demonstrate that their method yields results comparable to those obtained using a prior technique called Approximate Description Length (ADL) and elucidate the connections between the two methods.

**Strengths:**

1. The paper is mathematically solid and clearly presents its main ideas and results, making it easy to understand the concepts discussed.

2. The proposed technique, multicovers, is both novel and interesting, and it is shown to be a valuable tool for establishing tight sample complexity bounds for deep neural networks.

**Weaknesses:**

1. While the multicovers technique is shown to be effective for deriving tight sample complexity bounds for deep neural networks with smooth activation functions, as established by previous ADL methods, further discussion is needed on how this technique could enhance the exploration of sample complexity in areas of deep learning that have not been thoroughly investigated. This includes potential applications to other deep learning models and fields, and how multicovers might provide insights or advancements in those contexts.

2. There are some typos in the paper, and the authors should conduct a thorough review to correct these issues.

**Questions:**

See weakness.

---

### Official Review · Reviewer_D6gk · 2024-11-09

**Soundness:** 2
**Presentation:** 1
**Contribution:** 2
**Rating:** 5
**Confidence:** 2

**Summary:**

This paper focuses on the analysis of the sample complexity for the generalization of multi-layer neural networks with smooth activation functions.
The authors introduce a novel concept of multicover and derive sample complexity bounds by a careful estimate for multicover numbers.
The sample complexity for generalizaiton is derived by studying the uniform convergence property of neural networks, which is known to be a sufficient condition for the former property.

**Strengths:**

The introduction gradually introduces the main problem considered, providing even a concrete and meaningful example.
The main contribution consists of providing the concept of multicover and adopting it in the study of the sample complexity for uniform convergence.
The authors further draw connections with the related notion of Approximate Description Length (ADL), which is another successful notion for the same task.
These results provide additional insights with respect to the generalization property for the concept class of neural networks with smooth activations.

**Weaknesses:**

Considering related work, it appears that the sample complexity bound achieved in this work is essentially the same as the one achieved via ADL, up to polylog factors.
It also requires smoothness property that a relevant portion of prior work does not require.
The notion of multicover is nevertheless interesting in its own right, as it is more tightly related to the standard notion of covering adopted in classical uniform convergence results.

Moreover, the authors somewhat overlooked a thorough comparison with relevant prior work.
This causes a lack of contextualization (albeit only to some extent) for the results contained in this submission.
It also causes to miss the (still ongoing, apparently) debate about on the practical utility of characterizing the generalization for neural networks via uniform convergence.
The authors should at least provide some clarification about the discussions in, e.g., [1-4].

Unfortunately, the overall writing is lacking clarity and, in some parts, even detail.
It is often the case that some specific notation is used way before its actual definition (as in a different section).
Typos and misuse of the English language are also particularly frequent throughout the entire text.
Readers might easily be hindered by these shortcomings in the overall presentation, thus hindering a smooth and complete understanding of the results in this work.
Except for the introductory part, these issues are so radicated that fixing them would require a fundamental restructuring of the overall structure of the submitted paper (e.g., reordering sections and fixing typos seem unavoidable).

References:

[1] Nagarajan & Kolter. "Uniform convergence may be unable to explain generalization in deep learning". NeurIPS 2019.

[2] Neyshabur, Li, Bhojanapalli, LeCun, & Srebro. "The role of over-parametrization in generalization of neural networks". ICLR 2019.

[3] Negrea, Dziugaite, & Roy. "In defense of uniform convergence: Generalization via derandomization with an application to interpolating predictors". ICML 2020.

[4] Koehler, Zhou, Sutherland, & Srebro. "Uniform convergence of interpolators: Gaussian width, norm bounds and benign overfitting". NeurIPS 2021.

**Questions:**

- Please, address some of the main points raised above.
- Do you believe the concept of multicover (and similarly, that of ADL) has the potential to be useful beyond the concept class of multi-layer neural networks?
- Could the authors comment on how their results compare with those presented in [2]?

---

### Meta-Review · Area_Chair_8GvW · 2024-12-05

**Metareview:**

This paper introduces multicovers for estimating covering numbers of deep neural networks. However, there are several issues behind this paper. For example, this paper overlooked a thorough comparison with relevant prior work; writing is unclear with massive typos.
It is unclear how the new techniques will be useful to prove new results. The authors didn't provide rebuttal to address the reviewers' concerns. Therefore, the AC recommends to reject.

**Additional Comments On Reviewer Discussion:**

There is no author rebuttal posted. All of reviewers have the negative evaluation on this submission.

---

### Decision · Program_Chairs · 2025-01-22

Reject